# Multi-timescale Reinforcement Learning by Value Reconstruction

**Zhan Su** [1]  **Peixi Peng** [1 2]  **Xinyu Hu** [1]  **Cong Li** [1]  **Yisen Zhao** [1]  **Zhuojian Li** [1]  **Yonghong Tian** [1 2 3]  **Fanqi Shen** [4]

## Abstract

Most reinforcement learning (RL) baselines maximize future cumulative rewards with a fixed single discount factor, which limits their performance in complex sequential decision-making tasks due to a failure to balance short-term objectives and long-term planning. To address this issue, this paper focuses on a multi-timescale critic framework, where each component corresponds to a Q-value with a distinct discount factor. Two key improvements are proposed: (1) A Neural Reward Decoder reconstructs the reward sequence from multi-scale Q-values, with value and reward reconstruction losses enhancing Q-value estimation consistency; (2) A cross-attention-based Q-weight predictor adaptively adjusts Q-value weights via current observations to generate the final Q-value for policy optimization. Extensive experiments on DMControl and CARLA benchmarks demonstrate that our method significantly outperforms state-of-the-art (SOTA) baselines. Furthermore, we validate the framework's generalizability by integrating it with both off-policy (SAC, DrQ-v2) and on-policy (PPO) algorithms, achieving consistent performance gains.

## 1. Introduction

Reinforcement Learning (RL) aims to train agents to learn optimal policies via environmental interaction, maximizing the long-term cumulative reward: $\mathbb{E}_\pi[\sum_{t=0}^{\infty} \gamma^t r(s_t, a_t)]$. The discount factor $\gamma$ critically dictates future reward decay, steering the agent toward immediate returns (small $\gamma$) or long-term outcomes (large $\gamma$). Most RL baselines (Schul-

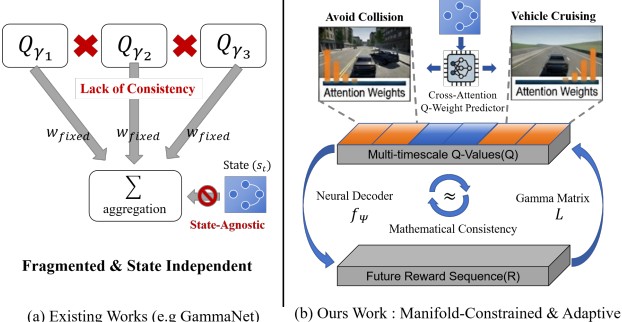

*Figure 1.* **(a) Conventional approaches** (e.g., Gamma-Nets (Sherstan et al., 2020), Laplace (Momennejad & Howard, 2018), Hyperbolic(Fedus et al., 2019)) typically rely on independent estimation and state-agnostic static aggregation. Crucially, they lack explicit consistency enforcement via the intrinsic mathematical constraint $(Q \leftrightarrow LR)$ between timescales. **(b) Our framework** introduces a closed-loop architecture that enforces manifold consistency via a Neural Reward Decoder and enables context-aware fusion through Cross-Attention, dynamically reweighting planning horizons based on state semantics (e.g., collision avoidance vs. cruising).

man et al., 2017; Haarnoja et al., 2018; Hessel et al., 2018; Gelada et al., 2019) adopt a fixed $\gamma$, confining decisions and evaluations to a single timescale. While effective for idealized environments (short horizons, simple rewards), this single-timescale setup fails to balance short-term objective pursuit and long-term strategy planning in complex sequential tasks.

Interestingly, recent advances in neuroscience have revealed that the brain's reinforcement learning mechanisms naturally incorporate multi-timescale representations. A seminal study published in Nature in 2025 (Masset et al., 2025) demonstrated that dopaminergic neurons in the brain do not employ a uniform discount factor when encoding reward prediction errors (RPE); instead, different dopaminergic neurons exhibit a diversity of discount time constants—some neurons are more tuned to immediate rewards, while others are sensitive to more distant reward signals. This heterogeneity at the neuronal level enables the brain to process reward information across different temporal spans in parallel. The study has validated the effectiveness of multi-timescale RL under relatively simple tasks. In addition, a few pioneering works have begun to integrate multi-timescale learning

[1]School of Electronic and Computer Engineering, Shenzhen Graduate School, Peking University, China [2]Pengcheng Laboratory, China [3]National Engineering Research Center of Visual Technology, School of Computer Science, Peking University, China [4]College of Computer Science and Technology, Zhejiang University, China. Correspondence to: Peixi Peng <pxpeng@pku.edu.cn>.

*Proceedings of the 43rd International Conference on Machine Learning*, Seoul, South Korea. PMLR 306, 2026. Copyright 2026 by the author(s).

mechanisms into RL algorithms. For instance, Fedus et al. (Fedus et al., 2019) utilized a single critic network that outputs multiple Q-value heads, with each head corresponding to a distinct discount factor $\gamma$. Similarly, Sherstan et al. (Sherstan et al., 2020) proposed an algorithm that takes a list of $\gamma$ values as parameters, calculates temporal-difference (TD) errors for each $\gamma$ independently, and then aggregates these errors to update a shared set of network parameters. Despite these efforts, current approaches ignore two important issues: Firstly, value estimations at different timescales are often conducted in a relatively independent manner, lacking an effective mechanism to explicitly establish intrinsic connections between estimates at different scales, which results in insufficient constraint for better value estimation; Secondly, an agent should consider the multi-timescale values comprehensively, and how to adaptively adjust the agent's focus on different timescales remains a challenge.

To achieve these objectives, we design two key enhancements for a multi-critic architecture where each critic corresponds to a Q-value with a distinct discount factor. For critic learning, building on the theoretical foundation from prior Nature work (Masset et al., 2025)—which demonstrates that the brain decodes reward information via multi-timescale value representations—we introduce a learnable neural decoder to approximate the inverse mapping and reconstruct reward sequences from multi-scale Q-value vectors. We further propose dual consistency losses: a value-reconstruction loss that integrates all multi-timescale values to establish intrinsic connections among their estimates, and an additional reward-reconstruction loss tailored to predict the first environment-collected reward—compensating for the unavailability of true values in off-policy settings. For actor learning, we develop a cross-attention-based *Q weight predictor* module to generate the final Q-value: leveraging the fact that different critics encode future characteristics across distinct timescales, we model observations as queries and multi-critic features as keys and values, enabling the cross-attention mechanism (Vaswani et al., 2017) to adaptively adjust the weights of individual Q-values by analyzing the relational semantics between current observations and multi-timescale future representations.

Notably inspired by prior biological work (Masset et al., 2025), our method differs in two key aspects: first, we supplement the value-reconstruction loss with a reward-reconstruction loss; second, we address the unexplored challenge of multi-Q fusion for policy optimization in (Masset et al., 2025) by proposing a cross-attention-based Q-weight predictor that explicitly integrates multi-critic outputs to guide actor updates. Designed as a plug-and-play module, our framework is compatible with any actor-critic RL algorithm, with core contributions summarized below:

(1) We propose a multi-$\gamma$ critic architecture that enables explicit multi-timescale value modeling for RL agents; (2) We introduce a neural reward decoder for inverse approximation with dual consistency losses to achieve reversible supervision between value estimates and reward sequences; (3) We design a cross-attention-based Q weight predictor that adaptively fuses value functions across temporal scales according to state-dependent information.

Extensive experiments on DMControl (Tassa et al., 2018) (including 6 common tasks and 5 hard tasks) and CARLA (Dosovitskiy et al., 2017) (under 4 weathers) demonstrate our method outperforms the baseline significantly and achieve SOTA performance.

## 2. Related Work

### 2.1. Multi-Timescale Reinforcement Learning

Multi-timescale reinforcement learning (MTRL) enhances an agent's capability to model decision-making across different temporal horizons by incorporating multiple discount factors ($\gamma$). Existing approaches include weighting multi-$\gamma$ returns to approximate hyperbolic discounting (Fedus et al., 2019), aggregating multiple policies trained under distinct $\gamma$ to approximate the average-reward optimal solution (Reinke et al., 2017), and treating $\gamma$ as a network input for unified modeling across continuous timescales (Sherstan et al., 2020). However, these methods lack an adaptive mechanism for selecting or emphasizing different timescales, limiting their ability to effectively fuse multi-timescale information.

### 2.2. Inverse Laplace Transform in RL

A central insight of multi-timescale learning is that value functions under different discount factors ($\gamma$) can be transformed into explicit future reward timing via the inverse Laplace transform. Momennejad et al. (Momennejad & Howard, 2018) first formalized this idea using multi-$\gamma$ successor representations, while Tano et al. (Tano et al., 2020) introduced a learnable inverse Laplace approximation based on local TD updates. However, these analytical or finite-difference approaches are sensitive to noise and numerically unstable under discrete $\gamma$. Masset et al. (Masset et al., 2025) further demonstrated that a neural decoder can robustly approximate the inverse Laplace transform even in noisy settings. Building on this line of work, we employ a learnable neural decoder for inverse Laplace approximation and, for the first time, use the decoded reward as an auxiliary loss to regularize critic learning, enabling stable end-to-end multi-timescale value modeling.

### 2.3. Multi-Critic Architectures and Q-value Fusion

A core challenge in multi-critic reinforcement learning is effectively integrating multiple value estimates to improve

policy stability and performance. Existing Q-value fusion methods include REDQ (Chen et al., 2021), which averages a random subset of critics; AdaEQ (Wang et al., 2021), which adaptively re-weights critics based on error feedback; CTD4 (Valencia et al., 2025), which applies Kalman filtering to fuse value distributions; and DEA (Werge et al., 2025), which adjusts critic conservatism through learnable directional parameters. However, these approaches are not designed for multi-timescale value estimation and lack state-dependent adaptive fusion mechanisms.

## 3. Preliminary

### 3.1. Reinforcement Learning

The core framework of RL is modeled as a Markov Decision Process (MDP) (BELLMAN, 1957), described by the tuple $(\mathcal{S}, \mathcal{A}, P, r, \gamma)$. Here, $\mathcal{S}$ and $\mathcal{A}$ denote the state and action spaces, $P(s'|s, a)$ is the transition probability, $r(s, a)$ is the reward function, and $\gamma \in [0, 1)$ is the discount factor. The agent aims to learn a policy $\pi$ that maximizes the expected cumulative discounted return $J(\pi) = \mathbb{E}_\pi[\sum_{t=0}^{\infty} \gamma^t r_t]$. We adopt the general Actor-Critic architecture, which serves as the foundation for various state-of-the-art algorithms, including off-policy methods (e.g., SAC (Haarnoja et al., 2018), DrQ-v2 (Yarats et al., 2021)) and on-policy methods (e.g., PPO (Schulman et al., 2017)). **Critics** (parameterized by $\theta$) aim to estimate the value function (either $Q^\pi(s, a)$ or $V^\pi(s)$) by minimizing the temporal difference (TD) error or regression loss against a bootstrapped target $y$:

$$\mathcal{L}_{\text{Critic}}(\theta) = \mathbb{E}\left[(\mathcal{V}_\theta - y)^2\right], \quad (1)$$

where $\mathcal{V}_\theta$ denotes the estimated value, and $y$ is the target constructed based on the specific algorithm (e.g., soft targets for SAC (Haarnoja et al., 2018) or standard Bellman targets for DrQ-v2 (Yarats et al., 2021)). **Actors** (parameterized by $\phi$) are updated to maximize the expected performance objective:

$$\mathcal{L}_\pi(\phi) = -\mathbb{E}\left[\mathcal{M}(s, a)\right]. \quad (2)$$

Here, $\mathcal{M}$ represents the optimization metric guiding the policy update. For off-policy algorithms, $\mathcal{M}$ maximizes Q-value objectives (e.g., $Q_\theta(s, \pi_\phi(s))$ for DrQ-v2, or entropy-augmented Q-values for SAC), while on-policy algorithms like PPO utilize advantage estimates, such as Generalized Advantage Estimation (GAE) (Schulman et al., 2015).

### 3.2. Multi-Timescale Value Representation

Standard RL relies on a single fixed discount factor $\gamma$, which assumes a fixed planning horizon (Haarnoja et al., 2018). To enable multi-timescale decision-making, we introduce a set of distinct discount factors $\Gamma = \{\gamma_1, \gamma_2, \ldots, \gamma_N\}$. For each $\gamma_i$, the corresponding value function is defined as:

$$Q_i(s, a) = \mathbb{E}_\pi\left[\sum_{t=0}^{\infty} \gamma_i^t r_t\right] \approx \mathbb{E}_\pi\left[\sum_{t=0}^{T-1} \gamma_i^t r_t\right], \quad (3)$$

where the approximation holds when the Decoding Horizon $T$ is sufficiently large such that the tail contribution $\gamma_i^T \to 0$.

**Remark.** The Decoding Horizon $T$ is a short, local sliding lookahead window strictly used for auxiliary reward reconstruction. It is mathematically decoupled from, and significantly shorter than, the environment's full episode length.

Let $\mathbf{R} = [r_0, r_1, \ldots, r_{T-1}]^\top \in \mathbb{R}^T$ be the finite reward sequence and $\mathbf{Q} = [Q_1, Q_2, \ldots, Q_N]^\top \in \mathbb{R}^N$ be the multi-scale value vector. The relationship between them allows for an approximate linear transformation:

$$\mathbf{Q} \approx \mathbf{LR}, \quad \text{where } \mathbf{L}_{i,t} = \gamma_i^t. \quad (4)$$

Here, $\mathbf{L} \in \mathbb{R}^{N \times T}$ is a Vandermonde matrix (Masset et al., 2025), which mathematically projects the temporal reward structure onto different discount scales.

**Ill-posedness of Analytical Inversion.** Analytically recovering $\mathbf{R} = \mathbf{L}^{-1}\mathbf{Q}$ is mathematically ill-posed due to the instability of the Vandermonde matrix $\mathbf{L}$.

**Proposition 3.1** (Ill-Conditioning of Multi-Scale Projection). *For a Vandermonde matrix $\mathbf{L}$ with distinct nodes $\gamma_i \in (0, 1)$, Gautschi (Gautschi, 1974) proved that the condition number is lower-bounded by $\kappa_\infty(\mathbf{L}) \geq C \cdot 2^N$, where $C \propto (\prod_{i \neq j} |\gamma_i - \gamma_j|)^{-1}$.*

**Remark.** In RL, the presence of closely spaced discount factors minimizes the separation term, causing the coefficient $C$ to explode. Specifically, with $N = 6$, our empirical condition number reaches $\approx 2.64 \times 10^5$ (implying a massive $C \approx 4.12 \times 10^3$). This extreme amplification of noise renders analytical inversion infeasible, necessitating a learning-based decoder.

## 4. Methodology

### 4.1. Overview

We propose a robust framework illustrated in Fig. 2. Our method extends the standard actor-critic architecture by (1) learning a Neural Reward Decoder to approximate the inverse mapping from multi-scale values to rewards, and (2) utilizing this structure to adaptively fuse critics for policy optimization. Formally, the ensemble produces a multi-scale value vector $\mathbf{V} \in \mathbb{R}^N$, representing state-action values $\mathbf{Q}_\theta(s, a)$ in off-policy settings (e.g., SAC (Haarnoja et al., 2018), DrQ-v2 (Yarats et al., 2021)) or state values $\mathbf{V}_\theta(s)$ in on-policy settings (e.g., PPO (Schulman et al., 2017)):

$$\mathbf{V}_\theta = [V_1, V_2, \ldots, V_N]^\top. \quad (5)$$

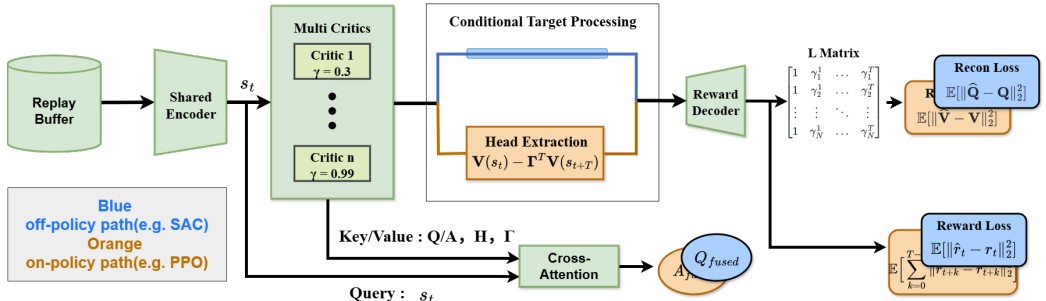

*Figure 2.* Framework overview. The critic is extended into a multi-timescale structure with multiple $\gamma$-specific critics and two auxiliary losses (reward loss and reconstruction loss). The actor remains unchanged, and Q-values are fused through cross-attention.

### 4.2. Neural Reward Decoder via Inverse Approximation

Instead of calculating the unstable analytical inverse $\mathbf{L}^{-1}$, we introduce a parameterized neural network $f_\psi$ to approximate the decoding process. This allows the model to learn a space of valid reward sequences. We formulate two distinct decoding strategies tailored to the data availability of different RL paradigms.

**Implicit Tail Marginalization (Off-Policy).** In off-policy settings (e.g., DrQ-v2 (Yarats et al., 2021)), the distributional mismatch between historical buffer trajectories and the current policy prevents utilizing stored future rewards for value decomposition. Consequently, we assume $\mathbf{V}_\theta(s_t) \approx \mathbf{L}\mathbf{R}_{0:T-1}$ by omitting negligible tail terms, allowing the decoder $f_\psi$ to infer the reward sequence directly from the current value vector:

$$\widehat{\mathbf{R}} = f_\psi(\mathbf{V}_\theta(s_t)). \tag{6}$$

**Explicit Head Value Extraction (On-Policy).** In contrast, on-policy algorithms (e.g., PPO (Schulman et al., 2017)) collect and update on complete sequential trajectories. This structure grants direct access to future states $s_{t+T}$ within the current rollout buffer. We leverage this availability to explicitly isolate the **Head Value** corresponding to the interval $[t, t+T]$ by subtracting the discounted tail value:

$$\mathbf{V}_{head,\theta} = \mathbf{V}_\theta(s_t) - \mathbf{\Gamma}^T \mathbf{V}_\theta(s_{t+T}), \tag{7}$$

where $\mathbf{\Gamma} = \mathrm{diag}(\gamma_1, \ldots, \gamma_N)$. Consequently, the decoder operates on this cleaner signal, free from long-term approximation errors:

$$\widehat{\mathbf{R}} = f_\psi(\mathbf{V}_{head,\theta}). \tag{8}$$

### 4.3. Bidirectional Consistency Losses

We introduce two complementary losses to enforce consistency between the latent value estimates and the physical reward structure, again adapting to the data constraints.

**Reward Reconstruction Loss.** We supervise the decoded rewards using ground-truth data. For **Off-Policy** methods, due to the distributional mismatch between historical behavioral policies in the replay buffer and the current target policy, supervising the entire decoded sequence introduces severe bias. Thus, only the immediate reward $r_t$ provides an unbiased signal for supervision:

$$\mathcal{L}_r(\xi, \theta, \psi) = \mathbb{E}\big[\|\hat{r}_t - r_t\|_2^2\big]. \tag{9}$$

For **On-Policy** methods, due to the alignment between the sample distribution and the current policy, we safely supervise the entire decoded sequence:

$$\mathcal{L}_r(\xi, \theta, \psi) = \mathbb{E}\big[ \sum_{k=0}^{T-1} \|\hat{r}_{t+k} - r_{t+k}\|_2^2\big]. \tag{10}$$

**Value Reconstruction Loss.** We project the decoded rewards back into the value space using the fixed forward matrix $\mathbf{L}$ to ensure cycle consistency. For **Off-Policy** methods, we reconstruct the full value vector:

$$\widehat{\mathbf{V}} = \mathbf{L}\widehat{\mathbf{R}}, \quad \mathcal{L}_{recon}(\xi, \theta, \psi) = \mathbb{E}\big[\|\widehat{\mathbf{V}} - \mathbf{V}_\theta(s_t)\|_2^2\big]. \tag{11}$$

For **On-Policy** methods, since the tail terms are explicitly removed in Eq. (7), we minimize the reconstruction error specifically for the head value:

$$\mathcal{L}_{recon}(\xi, \theta, \psi) = \mathbb{E}\big[\|\mathbf{L}\widehat{\mathbf{R}} - \mathbf{V}_{head,\theta}(s_t)\|_2^2\big]. \tag{12}$$

Here, $t$ denotes the exact environment timestep when the current state $s_t$ is sampled. To perfectly align with the environment's timeline, we index the predicted reward sequence generated by the Neural Reward Decoder as $\widehat{\mathbf{R}} = [\hat{r}_t, \hat{r}_{t+1}, \ldots, \hat{r}_{t+T-1}]^\top$, where $\hat{r}_{t+k}$ directly corresponds to the ground-truth reward $r_{t+k}$ at the future timestep $t+k$.

### 4.4. Adaptive Fusion via Cross-Attention

To synthesize a unified policy from diverse timescales, we employ a cross-attention module $g_\omega$ that adaptively computes weights $\boldsymbol{\alpha}(s)$ by analyzing the relational semantics

*Table 1.* We compare our method against standard SAC and PPO baselines. Mean $\pm$ std over 5 seeds.

| Task | SAC (Off-Policy) | | PPO (On-Policy) | |
|---|---|---|---|---|
| | Baseline | **Ours** | Baseline | **Ours** |
| Cartpole, Swingup | $330 \pm 73$ | $\mathbf{540 \pm 76}$ | $190 \pm 11$ | $\mathbf{432 \pm 89}$ |
| Reacher, Easy | $307 \pm 65$ | $\mathbf{572 \pm 61}$ | $310 \pm 16$ | $\mathbf{542 \pm 106}$ |
| Cheetah, Run | $85 \pm 51$ | $\mathbf{381 \pm 59}$ | $95 \pm 83$ | $\mathbf{209 \pm 69}$ |
| Walker, Walk | $71 \pm 52$ | $\mathbf{130 \pm 82}$ | $87 \pm 56$ | $\mathbf{251 \pm 210}$ |
| Finger, Spin | $346 \pm 95$ | $\mathbf{736 \pm 53}$ | $118 \pm 171$ | $\mathbf{520 \pm 57}$ |
| Ball in cup, Catch | $162 \pm 122$ | $\mathbf{492 \pm 48}$ | $206 \pm 5$ | $\mathbf{720 \pm 127}$ |
| *Average Score* | 206.8 | **475.2** | 167.7 | **445.7** |

between the current state and multi-scale future representations. Specifically, we use the state embedding $s$ as the query, while the keys and values are derived from a concatenated feature vector of each critic's intermediate latent feature $h_i(s)$, estimated value $V_i$, and discount factor $\gamma_i$. The attention weights are calculated as:

$$\boldsymbol{\alpha}(s) = \mathrm{softmax}\big(g_\omega(s, \mathrm{concat}(\{h_i(s), V_i, \gamma_i\}_{i=1}^N))\big). \quad (13)$$

The fusion strategy adapts to the RL paradigm (Q-values for off-policy, Generalized Advantage Estimation for on-policy):

$$\mathcal{E}_{\mathrm{fused}}(s, a) = \sum_{i=1}^N \alpha_i(s)\mathcal{E}_i(s, a), \quad (14)$$

where $\mathcal{E}_i$ represents either $Q_i(s, a)$ or $A_i(s, a)$. This mechanism allows the agent to dynamically prioritize different planning horizons based on the specific environmental context.

### 4.5. Complete Optimization Framework

The complete training procedure integrates multi-scale critic learning, reward reconstruction, and adaptive value fusion. Algorithm 1 summarizes the main steps, which can be seamlessly integrated into various RL baselines.

## 5. Experiments

The experiments are conducted on DMControl (Tassa et al., 2018) and CARLA (Dosovitskiy et al., 2017) benchmarks (see Fig. 3). Based on the experimental design, we aim to validate three key claims in the following order: (1) **Universality**: our method serves as a generic plugin that consistently improves various RL paradigms (e.g., SAC (Haarnoja et al., 2018), PPO (Schulman et al., 2017)); (2) **SOTA Performance**: when integrated with a strong baseline (DrQ-v2 (Yarats et al., 2021)), our method achieves state-of-the-art performance compared to existing visual control and multi-timescale baselines; (3) **Effectiveness**: rigorous ablations and visualizations confirm the necessity of each component and the interpretability of our mechanism. Additionally, extended empirical analyses are provided in the Appendix.

**Algorithm 1** Unified Multi-Timescale Critic Framework Training

1: **Input:** Shared Encoder $E_\xi$, Critics $\mathbf{V}_\theta$, Decoder $f_\psi$, Actor $\pi_\phi$, Attention $g_\omega$, Discount factors $\{\gamma_i\}_{i=1}^N$.
2: **Input:** Experience Dataset $\mathcal{D}$ (Replay Buffer or Trajectory Buffer).
3: Initialize parameters $\xi, \theta, \psi, \phi, \omega$ randomly
4: **repeat**
5:     Sample data batch $\mathcal{B}$ from $\mathcal{D}$
6:     Compute latent representation $s = E_\xi(o)$ from observation $o$
7:     /* 1. Optimization of Representation & Evaluation */
8:     Compute V vector $\mathbf{V}_\theta(s) = [V_1(s), \ldots, V_N(s)]^\top$
9:     Calculate critic loss $\mathcal{L}_{\mathrm{critic}} = \sum \mathrm{MSE}(\mathbf{V}_\theta, \mathbf{V}_{\mathrm{target}})$
10:     **Construct Target Head Value** $\mathbf{V}_{head,\theta}$:
11:       ▷ *Off-policy:* $\mathbf{V}_{head,\theta} \leftarrow \mathbf{V}_\theta(s)$
12:       ▷ *On-policy:* Compute $\mathbf{V}_{head,\theta}$ via Eq. 7
13:     Decode reward sequence $\widehat{\mathbf{R}}$ via Eq. 6 or Eq. 8
14:     Calculate losses $\mathcal{L}_r$ and $\mathcal{L}_{\mathrm{recon}}$ via Eq. 9-12
15:     **Joint Update:** Minimize $\mathcal{L}_{total} = \mathcal{L}_{\mathrm{critic}} + \mathcal{L}_r + \mathcal{L}_{\mathrm{recon}}$
16:     w.r.t parameters $\xi, \theta, \psi$
17:     /* 2. Adaptive Fusion & Policy Optimization */
18:     Compute mixing weights $\boldsymbol{\alpha}(s)$ via Eq. 13
19:     **Determine Evaluation Metric** $\mathcal{E}$ for fusion:
20:       ▷ *Off-policy:* $\mathcal{E}_i \leftarrow Q_{\theta_i}(s, a)$
21:       ▷ *On-policy:* $\mathcal{E}_i \leftarrow A_i(s, a)$
22:     Compute fused estimate $\mathcal{E}_{\mathrm{fused}}$ via Eq. 14
23:     **Joint Update:** Maximize $\mathcal{E}_{\mathrm{fused}}$ w.r.t parameters $\xi, \phi, \omega$
24:     Update target networks if applicable
25: **until** training converges

### 5.1. Experiment Setup

In our multi-timescale configuration, we employ $N = 6$ critics with discount factors $\gamma \in \{0.3, 0.5, 0.7, 0.8, 0.9, 0.95\}$ and adopt a reward decoding length $T$ of 15. Our framework is designed as a plug-and-play module. To evaluate generalizability, we apply our method to SAC (Haarnoja et al., 2018) and PPO (Schulman et al., 2017). For the primary evaluation, we integrate it into DrQ-v2 (Yarats et al., 2021), with all experiments conducted without modifying the original model architectures or hyperparameter settings.

The experimental settings on DMControl follow prior works (Laskin et al., 2020; Zheng et al., 2023; Grooten et al., 2023; Liu et al., 2025), covering 6 common tasks and 5 hard tasks. For CARLA, we select the highway driving task under diverse weather conditions (Ma et al., 2023). The evaluation metric and reward design largely follow DBC (Zhang et al., 2020), where the agent aims to drive as far as possible along a figure-eight highway within 2000 timesteps without collisions. All experiments are conducted

*Table 2.* Performance comparison with SOTA methods on DMControl 6 common tasks at 500K environment steps. Mean ± standard deviation over 5 seeds. Bold numbers indicate the best performance. And DrQ-v2 is our primary baseline.

| 500K STEP SCORES | CURL ICML'20 | SVEA NeurIPS'21 | MLR NeurIPS'22 | PSRL CVPR'23 | TACO NeurIPS'23 | MaDi AAMAS'24 | Resact AAAI'25 | DrQ-v2 ArXiV'21 | Ours This work |
|---|---|---|---|---|---|---|---|---|---|
| Cartpole, Swingup | 841 ± 45 | 865 ± 10 | 872 ± 5 | **895 ± 39** | 870 ± 21 | 849 ± 6 | 870 ± 12 | 845 ± 18 | 857 ± 3 |
| Reacher, Easy | 929 ± 44 | 944 ± 52 | 957 ± 41 | 932 ± 41 | 944 ± 50 | 955 ± 31 | 974 ± 16 | 971 ± 4 | **980 ± 3** |
| Cheetah, Run | 518 ± 28 | 682 ± 65 | 674 ± 37 | 686 ± 80 | 663 ± 30 | 732 ± 45 | 750 ± 8 | 700 ± 29 | **756 ± 20** |
| Walker, Walk | 902 ± 43 | 919 ± 24 | 939 ± 10 | 930 ± 75 | 914 ± 87 | 912 ± 26 | 953 ± 21 | 922 ± 70 | **958 ± 7** |
| Finger, Spin | 926 ± 45 | 924 ± 93 | 973 ± 31 | 961 ± 121 | 972 ± 89 | 951 ± 47 | **979 ± 4** | 947 ± 45 | 977 ± 7 |
| Ball in cup, Catch | 959 ± 27 | 960 ± 19 | 964 ± 14 | **988 ± 54** | 960 ± 22 | 912 ± 62 | 967 ± 4 | 943 ± 186 | 982 ± 4 |
| Average | 845.8 | 882.3 | 896.5 | 894.1 | 887.1 | 885.1 | 915.5 | 888.0 | **918.3** |

*Table 3.* Performance comparison with SOTA methods on DMControl benchmarks 5 hard tasks (500K Environment Steps). Mean ± standard deviation over 5 seeds. Bold numbers indicate the best performance. And DrQ-v2 is our primary baseline.

| 500K STEP SCORES | Flare NIPS'21 | TACO NeurIPS'23 | MaDi AAMAS'24 | Resact AAAI'25 | DrQ-v2 ArXiV'21 | Ours This work |
|---|---|---|---|---|---|---|
| Quadruped, Walk | 296 ± 139 | 345 ± 89 | 277 ± 92 | 385 ± 81 | 343 ± 106 | **734 ± 31** |
| Pendulum, Swingup | 242 ± 152 | 485 ± 167 | 80 ± 24 | 618 ± 380 | 840 ± 27 | **858 ± 6** |
| Hopper, Hop | 90 ± 55 | 112 ± 42 | 80 ± 24 | 99 ± 49 | 116 ± 94 | **120 ± 28** |
| Finger, Turn Hard | 282 ± 67 | 372 ± 174 | 311 ± 143 | **465 ± 153** | 138 ± 95 | 421 ± 122 |
| Walker, Run | 426 ± 33 | 355 ± 89 | 382 ± 87 | 467 ± 27 | 436 ± 143 | **509 ± 43** |
| Average | 267.2 | 333.8 | 284.4 | 374.6 | 406.8 | **528.4** |

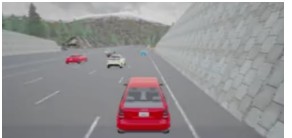
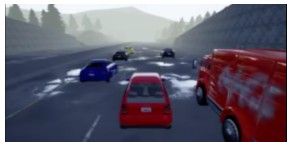

*(a)* CARLA

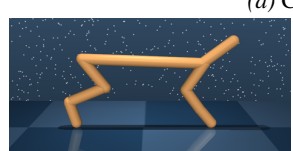
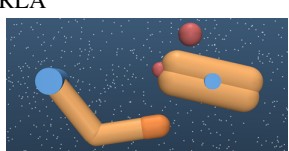

*(b)* DMControl

*Figure 3.* Examples of experiment simulators.

over 5 seeds, reporting the mean rewards and standard deviations. Detailed settings are provided in the supplementary material.

## 5.2. Generalizability across RL Paradigms

To demonstrate that our method is a generic plugin, we integrated it into both off-policy (SAC (Haarnoja et al., 2018)) and on-policy (PPO (Schulman et al., 2017)) frameworks. As shown in Table 1, our method consistently improves the performance of both baselines.

## 5.3. Comparison with State-of-the-Art Methods

**DMControl Performance.** For 6 common tasks, we compare our method against 8 competitive baselines, including CURL (Laskin et al., 2020), DrQ-v2 (Yarats et al., 2021), MLR (Yu et al., 2022), PSRL (Choi et al., 2023), TACO (Zheng et al., 2023), MaDi (Grooten et al., 2023), and Resact (Liu et al., 2025), with baseline results derived from prior literature (Liu et al., 2025; Hafner et al., 2025). As shown in Table 2, our method achieves the best performance in 3 tasks, competitive results in the rest, and superior performance in terms of the aggregate average score.

Since common DMControl tasks are approaching performance saturation, we further evaluate on 5 hard tasks selected by Flare (Shang et al., 2021). Evaluations at 500k and 1M environment steps (Tables 3 and 4) show our method outperforms existing methods in almost all hard tasks, which involve partial observability, sparse rewards, and high-precision control requirements—highlighting the robustness of our approach in complex scenarios.

In summary, our method consistently outperforms the strong baseline DrQ-v2 across all tasks and achieves SOTA performance comparable to or exceeding advanced model-based methods like Dreamer-v3, with particularly significant advantages in hard tasks. Notably, while the margin over baselines like Resact is narrower on easier tasks due to performance saturation, our method is orthogonal to their design, suggesting potential for complementary integration. Its substantial lead on hard tasks further underscores the

*Table 4.* Performance comparison with SOTA methods on DMControl benchmarks 5 hard tasks (1M Environment Steps). Mean $\pm$ standard deviation over 5 seeds. Bold numbers indicate the best performance. The Dreamer-v3 method did not provide the corresponding standard deviation data, so it is not listed here. And DrQ-v2 is our primary baseline.

| 1M STEP SCORES | Flare NIPS'21 | TACO NeurIPS'23 | MaDi AAMAS'24 | Resact AAAI'25 | Dreamer-v3 Nature'25 | DrQ-v2 ArXiV'21 | Ours This work |
|---|---|---|---|---|---|---|---|
| Quadruped, Walk | $488 \pm 221$ | $665 \pm 144$ | $621 \pm 172$ | $690 \pm 128$ | $811 \pm -$ | $732 \pm 91$ | $\mathbf{846 \pm 25}$ |
| Pendulum, Swingup | $809 \pm 31$ | $784 \pm 42$ | $751 \pm 41$ | $817 \pm 6$ | $744 \pm -$ | $838 \pm 38$ | $\mathbf{871 \pm 9}$ |
| Hopper, Hop | $217 \pm 59$ | $221 \pm 45$ | $201 \pm 43$ | $233 \pm 32$ | $227 \pm -$ | $221 \pm 85$ | $\mathbf{236 \pm 8}$ |
| Finger, Turn Hard | $661 \pm 315$ | $672 \pm 167$ | $695 \pm 133$ | $857 \pm 80$ | $\mathbf{904 \pm -}$ | $491 \pm 182$ | $759 \pm 112$ |
| Walker, Run | $556 \pm 93$ | $582 \pm 63$ | $562 \pm 68$ | $554 \pm 21$ | $684 \pm -$ | $538 \pm 115$ | $\mathbf{721 \pm 9}$ |
| **Average** | 546.2 | 584.8 | 566.0 | 630.2 | 674 | 564.0 | **686.6** |

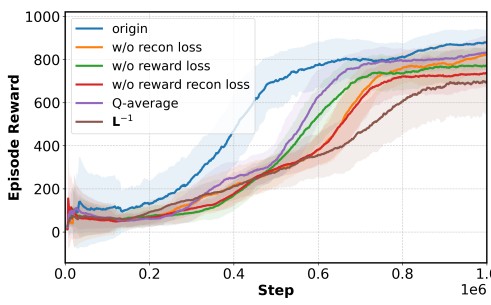

*(a)* Quadruped Walk in DMControl.

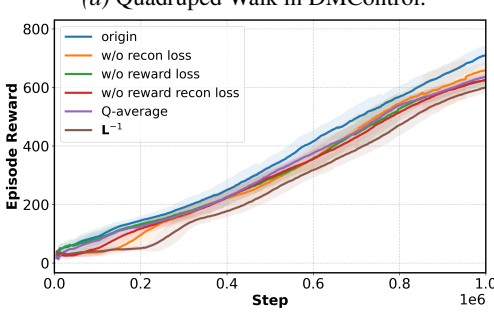

*(b)* Walker Run in DMControl.

*Figure 4.* Learning curves for different ablations.

specific efficacy of our approach in handling complex, long-horizon control challenges.

**CARLA Autonomous Driving.** The CARLA simulator features visual observations with substantial task-irrelevant information, serving as a rigorous testbed for realism. Results are compared against previous works (Liu et al., 2025; Ma et al., 2023). Table 6 indicates that our method achieves a significant improvement in episodic reward compared to baselines under various weather conditions, further validating the robustness of our visual representations.

**5.4. Comparison with Multi-Timescale Baselines**

We further validate our multi-timescale mechanism by comparing it with three relevant approaches: **Gamma-Nets** (Sherstan et al., 2020) (single critic with $\gamma$ input),

*Table 5.* Sensitivity analysis of the critic count ($N$) and Decoding Horizon ($T$). The baseline (DrQ-v2) relies on a single scale and does not utilize $T$. Results are reported as mean $\pm$ std at 1M steps over 5 seeds.

**(a) Impact of Critic Count $N$ (Ours fixed at $T = 15$)**

| Method | $\gamma$ List | Quadruped Walk | Walker Run |
|---|---|---|---|
| DrQ-v2 (Baseline) | Single | $732 \pm 91$ | $538 \pm 115$ |
| Ours ($N = 2$) | Random | $797 \pm 34$ | $649 \pm 14$ |
| Ours ($N = 4$) | Random | $830 \pm 22$ | $717 \pm 23$ |
| Ours ($N = 6$) | Random | $827 \pm 53$ | $725 \pm 5$ |
| Ours ($N = 8$) | Random | $831 \pm 15$ | $726 \pm 12$ |

**(b) Impact of Decoding Horizon $T$ (Ours fixed at $N = 6$)**

| Method | Horizon | Quadruped Walk | Walker Run |
|---|---|---|---|
| DrQ-v2 (Baseline) | N/A | $732 \pm 91$ | $538 \pm 115$ |
| Ours ($T = 10$) | 10 steps | $850 \pm 42$ | $721 \pm 18$ |
| Ours ($T = 15$) | 15 steps | $846 \pm 25$ | $721 \pm 9$ |
| Ours ($T = 20$) | 20 steps | $857 \pm 31$ | $714 \pm 12$ |

**Laplace** (Momennejad & Howard, 2018) (analytical inverse Laplace transforms), and **Hyperbolic** (Fedus et al., 2019) (fixed hyperbolic weighting). Since these methods were designed for different baselines, we re-implemented their core components in our DrQ-v2 framework for fairness (implementation details in the supplementary).

Experimental results on 5 DMControl hard tasks (Fig. 5) show our method outperforms these alternatives, attributed to three key advantages: (1) Unlike Gamma-Nets, we explicitly model the timescale relationship ($Q \approx LR$) for joint learning; (2) Unlike Laplace, our learnable decoder avoids numerical errors from analytical inverse approximations; (3) Unlike Hyperbolic's fixed weights, our cross-attention adaptively fuses Q-values based on the current state.

**5.5. Ablation Studies**

**Component Analysis.** To validate each module, we conducted 5 ablation experiments on the Quadruped Walk (DM-Control) and CARLA tasks: **w/o recon loss** (removing $\mathcal{L}_{\text{recon}}$), **w/o reward loss** (removing $\mathcal{L}_r$), **w/o both losses**,

*Table 6.* Comparison of average episode scores under different weather settings in CARLA (200k training Steps). Mean $\pm$ standard deviation over 3 seeds. Best results are in **bold**. And DrQ-v2 is our primary baseline.

| **Weather** | CycAug NIPS'23 | TACO NIPS'23 | MaDi AAMAS'24 | ResAct AAAI'25 | DrQ-v2 ArXiV'21 | Ours This work |
|---|---|---|---|---|---|---|
| Default | $263.9 \pm 12$ | $258.0 \pm 23$ | $227.0 \pm 19$ | $293.0 \pm 25$ | $221.7 \pm 34$ | $\mathbf{338.0} \pm 15$ |
| WetNoon | $277.6 \pm 9$ | $270.1 \pm 12$ | $241.8 \pm 12$ | $294.9 \pm 12$ | $216.8 \pm 12$ | $\mathbf{345.5} \pm 19$ |
| SoftRainNoon | $281.7 \pm 13$ | $260.6 \pm 12$ | $233.4 \pm 12$ | $280.1 \pm 15$ | $223.9 \pm 21$ | $\mathbf{354.6} \pm 16$ |
| HardRainSunset | $273.1 \pm 15$ | $273.4 \pm 12$ | $225.9 \pm 12$ | $292.3 \pm 19$ | $231.9 \pm 7$ | $\mathbf{334.3} \pm 18$ |
| **Average Score** | 274.1 | 265.5 | 232.0 | 290.1 | 223.6 | **343.1** |

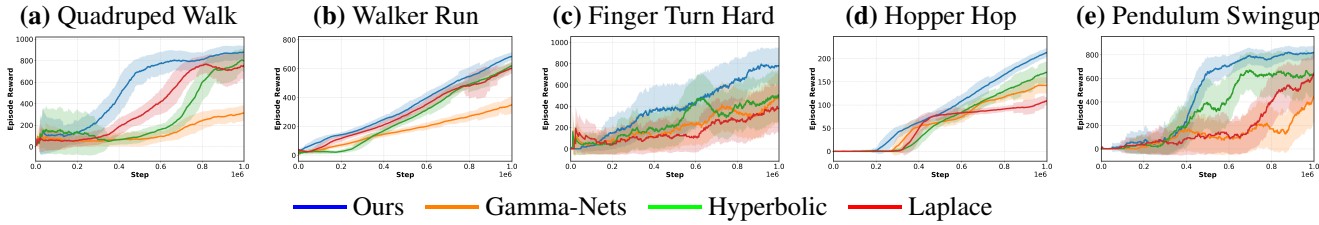

**(a)** Quadruped Walk  **(b)** Walker Run  **(c)** Finger Turn Hard  **(d)** Hopper Hop  **(e)** Pendulum Swingup

Ours — Gamma-Nets — Hyperbolic — Laplace

*Figure 5.* Comparison of different multi-timescale mechanism across 5 DMControl hard tasks, under 5 random seeds.

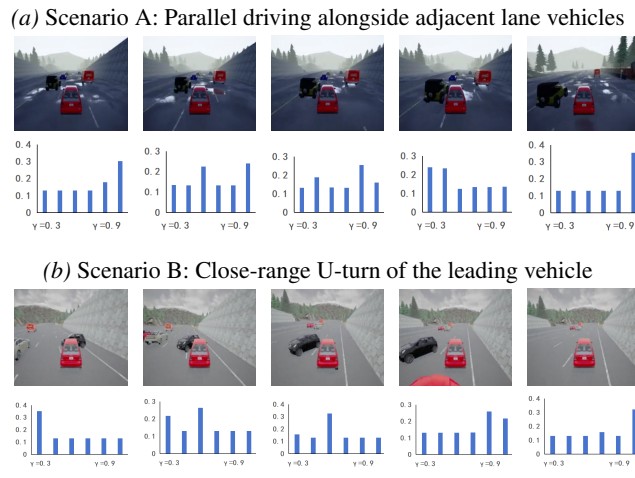

*(a)* Scenario A: Parallel driving alongside adjacent lane vehicles

*(b)* Scenario B: Close-range U-turn of the leading vehicle

*Figure 6.* Visualization of Q-weight predictor dynamics in two representative CARLA driving scenarios.

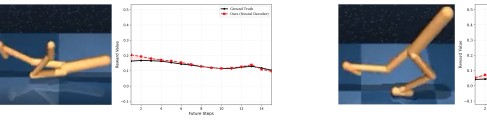

*(a)* Fallen Posture Reconstruction.

*(b)* Locomotion Posture Reconstruction.

*Figure 7.* **Reward reconstruction fidelity across dynamic regimes.** We visualize the decoding performance in (a) Fallen and (b) Locomotion poses on the DMControl Walker-walk task.

**Q-average** (replacing the attention fusion with arithmetic averaging), and $\mathbf{L}^{-1}$ (replacing the neural decoder with the analytical inverse $L^{-1}$ from (Masset et al., 2025)). As shown in Fig. 4, the learning curves demonstrate that every component is essential for optimal performance.

**Sensitivity to Structural Hyperparameters.** Our framework relies on two primary structural hyperparameters: the critic count $N$ and the Decoding Horizon $T$. To evaluate their impact, we conducted a sensitivity analysis (Table 5), comparing against the single-scale DrQ-v2 baseline (which does not utilize $T$). **Impact of $N$:** To replace manual tuning with a robust evaluation, we tested random $\gamma$ config-

urations. Specifically, across 5 trials, we generated initial lists comprising a fixed maximum discount of 0.95 and 7 random values in $(0, 0.95)$ to evaluate $N = 8$. We then iteratively dropped two random non-0.95 values to evaluate $N \in \{6, 4, 2\}$. As shown in Table 5a, any multi-scale configuration ($N \geq 2$) significantly outperforms the baseline. Performance improves sharply from $N = 2$ to $N = 4$ and plateaus thereafter, demonstrating that the framework is highly robust to the exact choice of $\gamma$ values once sufficient temporal resolution is reached. **Impact of $T$:** Table 5b illustrates the performance across different Decoding Horizons $T \in \{10, 15, 20\}$. The marginal variability across these settings confirms that $T$, as a local sliding lookahead window, provides stable auxiliary supervision without requiring meticulous tuning.

**Visualization.** Figure 6 visualizes the attention weights in CARLA. The model adaptively shifts focus between short- and long-horizon values depending on the driving context (e.g., focusing on short-term $\gamma$ when collision risk is high), providing interpretability for the fusion mechanism. Figure 7 visualizes the reconstructed reward signals against the

ground truth in the DMControl Walker-walk environment, demonstrating that our reward decoder achieves high-fidelity capture of the underlying reward manifold across diverse agent postures, including both stable and volatile regimes. This confirms the superiority of our neural approach over analytical inversion methods.

*Table 7.* Analysis of computational complexity and performance. Results are reported as mean $\pm$ standard deviation over 5 seeds.

| Method | Params | Train FLOPs | Quadruped Walk | | Walker Run | |
|---|---|---|---|---|---|---|
| | | | 500k | 1M | 500k | 1M |
| DrQ-v2 (Base) | 7M | 404.4M | 343 ± 106 | 732 ± 91 | 436 ± 143 | 538 ± 115 |
| DrQ-v2 (Widened) | 69M | 974.7M | 387 ± 71 | 760 ± 82 | 477 ± 62 | 558 ± 67 |
| Ours (Shared Trunk) | 18M | 363.7M | 726 ± 46 | 830 ± 54 | 496 ± 24 | **722** ± 10 |
| Ours (Independent) | 69M | 863.6M | **734** ± 31 | **846** ± 25 | **509** ± 43 | 721 ± 9 |

### 5.6. Computational Complexity and Efficiency

**Parameter Efficiency and Model Capacity.** To strictly isolate architectural overhead, we report Train FLOPs as the total theoretical compute—including all corresponding forward and backward passes—for a single complete RL update step per sample. To verify that performance gains stem from the multi-timescale design rather than increased capacity or compute, we evaluated a *Widened DrQ-v2* baseline by increasing its critic's hidden dimension from 1024 to 4608 to strictly match our maximum parameter budget. As shown in Table 7, this over-parameterization yielded only marginal gains. Subsequently, we evaluated a compact *Shared Trunk* implementation that: (1) shares the visual trunk across all critics; (2) shares the first two MLP layers as a unified Q-trunk; and (3) utilizes this shared trunk as the attention query. This significantly reduces Params and FLOPs while preserving SOTA performance, confirming that the framework's efficacy is driven by the multi-timescale mechanism rather than computational volume.

**Zero Inference Overhead.** The auxiliary components, including multi-scale critics, decoders, and the attention module, are strictly confined to the training phase. During deployment, the actor operates independently without these modules. This ensures that our framework incurs zero inference penalty, maintaining the identical real-time control efficiency as the standard baseline.

## 6. Conclusion

This paper proposes a multi-discount-factor RL framework that better models multi-timescale future returns, allowing the agent to jointly account for both short-term objectives and long-term strategies. The method achieves consistent improvements over strong baselines across tasks. Future work will focus on improving the framework's parameter efficiency, potentially through network distillation to reduce the training overhead without compromising performance.

## Acknowledgments

The study was funded by the Shenzhen Science and Technology Program (KQTD20240729102051063), the National Natural Science Foundation of China under contracts No. 62422602, No. 62372010, No. 62425101, No. 62332002, No. 62372010, No. 62206281, Key Laboratory Grants 241-HF-D05-01, and the major key project of the Peng Cheng Laboratory (PCL2021A13 and PCL2025A02). Computing support was provided by Pengcheng Cloudbrain.

## Impact Statement

This paper presents work whose goal is to advance the field of Machine Learning. There are many potential societal consequences of our work, none which we feel must be specifically highlighted here.

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

# A. Detailed Experimental Setup

*Table 8.* Hyper-parameters for DMControl experiments.

| Algorithms Hyper-parameters | |
| --- | --- |
| Replay buffer capacity | $10^6$ |
| Action repeat | 2 |
| Seed frames | 4000 |
| Exploration steps | 2000 |
| $n$-step returns | 3 |
| Mini-batch size | 256 |
| Optimizer | Adam |
| Learning rate | $10^{-4}$ |
| Agent update frequency | 2 |
| Critic Q-function soft-update rate $\tau$ | 0.01 |
| Features dim. | 50 |
| Repr. dim. | $32 \times 35 \times 35$ |
| Hidden dim. | 1024 |
| Exploration stddev. clip | 0.3 |
| Exploration stddev. schedule | easy: linear(1.0, 0.1, 100000) |
| | medium: linear(1.0, 0.1, 500000) |
| | hard: linear(1.0, 0.1, 2000000) |
| **Multi-$\gamma$ Module Hyper-parameters** | |
| $\gamma$ list | [0.3, 0.5, 0.7, 0.8, 0.9, 0.95] |
| Reward length | 15 |
| Critic number | 6 |
| Attention head | 4 |
| Reward decoder hidden unit | 128 |

In this section, we provide comprehensive implementation details of our experimental setup for evaluating the performance in both DMControl and CARLA environments. This includes the introduction of benchmarks, network architecture, and hyperparameters.

## A.1. Setup of DeepMind Control Suite

**Benchmarks:** We conduct evaluations on 6 common and 5 hard continuous control tasks, following the task setting of Flare (Shang et al., 2021). These tasks cover different types of tasks and various challenging elements. A detailed description is presented in Table 10.

**Hyper-parameters:** To demonstrate the general applicability of our method, we keep all environment-specific hyper-parameters from DrQ-v2 (Yarats et al., 2021) unchanged. On top of this base configuration, we introduce our components, whose hyper-parameters are summarized in Table 8.

## A.2. Setup of CARLA Simulator

**Benchmarks:** We adopt a modified version of DrQ-v2 based on the Rand PR method (Ma et al., 2023), enabling DrQ-v2 to operate in the CARLA environment. The reward function is defined as Eq. (15) from DBC (Zhang et al., 2020):

$$r_t = \mathbf{v}_{\text{ego}}^\top \hat{\mathbf{u}}_{\text{highway}} \cdot \Delta t - \lambda_i \cdot \text{impulse} - \lambda_s \cdot |\text{steer}|, \tag{15}$$

where $\mathbf{v}_{\text{ego}}$ represents the velocity vector of the ego vehicle projected onto the unit vector $\hat{\mathbf{u}}_{\text{highway}}$ that aligns with the highway direction. It is multiplied by the time discretization $\Delta t = 0.05$ to measure the progress of the vehicle on the highway in meters. Collisions are quantified in terms of impulses, measured in Newton-seconds. Additionally, a steering penalty is imposed, with steer $\in [-1, 1]$. The reward function incorporates weights $\lambda_i = 10^{-4}$ and $\lambda_s = 1$.

**Hyper-parameters:** To demonstrate the general applicability of our approach, we keep all hyperparameters and training configurations of the original method (Ma et al., 2023) unchanged, and implement our multi-timescale reinforcement learning framework on top of this baseline in Table 9.

*Table 9.* Hyper-parameters for CARLA experiments.

| Algorithms Hyper-parameters | |
| --- | --- |
| Replay buffer capacity | $10^5$ |
| Action repeat | 4 |
| Seed frames | 400 |
| Exploration steps | 100 |
| $n$-step returns | 3 |
| Mini-batch size | 512 |
| Optimizer | Adam |
| Learning rate | $10^{-4}$ |
| Agent update frequency | 2 |
| Critic Q-function soft-update rate $\tau$ | 0.01 |
| Features dim. | 50 |
| Repr. dim. | $32 \times 35 \times 119$ |
| Hidden dim. | 1024 |
| Exploration stddev. clip | 0.3 |
| Exploration stddev. schedule | linear(1.0, 0.1, 100000) |
| **Multi-$\gamma$ Module Hyper-parameters** | |
| $\gamma$ list | [0.3, 0.5, 0.7, 0.8, 0.9, 0.95] |
| Reward length | 15 |
| Critic number | 6 |
| Attention head | 4 |
| Reward decoder hidden unit | 128 |

*Table 10.* A detailed description of each task in our *common*, *hard* benchmarks.

| Task | Traits | Difficulty | Allowed Steps | $\dim(\mathcal{S})$ | $\dim(\mathcal{A})$ |
| --- | --- | --- | --- | --- | --- |
| Cartpole Swingup | swing, dense | common | $1 \times 10^6$ | 4 | 1 |
| Cup Catch | swing, catch, sparse | common | $1 \times 10^6$ | 8 | 2 |
| Cheetah Run | run, dense | common | $3 \times 10^6$ | 18 | 6 |
| Finger Spin | rotate, dense | common | $1 \times 10^6$ | 6 | 2 |
| Reacher Easy | reach, dense | common | $3 \times 10^6$ | 4 | 2 |
| Walker Walk | walk, dense | common | $1 \times 10^6$ | 18 | 6 |
| Finger Turn Hard | turn, sparse | hard | $3 \times 10^6$ | 6 | 2 |
| Hopper Hop | move, dense | hard | $3 \times 10^6$ | 14 | 4 |
| Pendulum Swingup | swing, sparse | hard | $1 \times 10^6$ | 2 | 1 |
| Quadruped Walk | walk, dense | hard | $3 \times 10^6$ | 56 | 12 |
| Walker Run | run, dense | hard | $3 \times 10^6$ | 18 | 6 |

### A.3. Setup of Multi-timescale Mechanism

We compare the proposed multi-timescale mechanism with three existing multi-timescale reinforcement learning methods: Gamma-Nets (Sherstan et al., 2020), Laplace RL (Momennejad & Howard, 2018), and Hyperbolic RL (Fedus et al., 2019). Since these methods were originally developed on different RL frameworks and evaluated in different environments, a direct and fully rigorous comparison is not feasible. To enable a reasonably fair and referenceable comparison, we re-implemented the core multi-timescale components of these methods within the DrQ-v2 framework. In this section we provide additional implementation details. **Gamma-Nets:** The core idea of Gamma-Nets is to parameterize the discount factor $\gamma$ and feed it into the critic as an additional input, enabling a single value function to learn across multiple temporal scales. Within our DrQ-v2 framework, we faithfully reproduce the sampling procedure described in the original paper: each update samples $N$ discount factors, includeing two boundary discounts $\{\gamma_{\min}, \gamma_{\max}\}$, while the remaining $\gamma$ values are generated by sampling half uniformly from the interval $[\gamma_{\min}, \gamma_{\max}]$ and sampling the other half uniformly in the timescale domain $\tau \in [\tau_{\min}, \tau_{\max}]$, followed by the conversion $\gamma = 1 - \frac{1}{\tau}$. All sampled discount factors are concatenated with the state and action inputs and passed through the shared critic, with standard TD updates performed independently for each $\gamma$. Moreover, the original paper notes that TD errors across different timescales differ significantly in magnitude, requiring a scaling mechanism to stabilize training. Based on the $n$-step derivation, each TD term associated with discount $\gamma_k$ should be multiplied by a common factor $(1 - \gamma_k)$ to normalize losses across timescales. Our implementation follows this prescription exactly.The key hyper-parameters specific to our implementation are summarized in Table 11.

**Laplace:** The core idea of this method (Momennejad & Howard, 2018) is to recover the future state sequence from multi-scale successor representations using an analytic Laplace inverse transform. The central operator is Eq.(16)

$$L_k^{-1} = C_k \, \sigma^{k+1} \frac{d^k}{d\sigma^k}, \quad \sigma = -\log\gamma, \; C_k = \frac{1}{k!}. \tag{16}$$

Following the analytic procedure described in the original paper, we apply this operator to a vector of Q-values across different discount factors to directly decode the immediate reward as in Eq.(17)

$$R \approx L_k^{-1} Q. \tag{17}$$

The key hyper-parameters specific to our implementation are summarized in Table 11.

**Hyperbolic:** In the original paper, hyperbolic discounting can be expressed as a weighted mixture of exponentially discounted value functions $Q^{\gamma_i}(s,a)$ as Eq.18

$$Q(s,a) \approx \sum_{i=1}^{N} (\gamma_{i+1} - \gamma_i) \frac{1}{k} \gamma_i^{\frac{1}{k}-1} Q^{\gamma_i}(s,a). \tag{18}$$

The sampling strategy of $\{\gamma_i\}$ is given by Eq.(19):

$$b = \exp\left( \frac{\log(1 - \gamma_{\max}^{1/k})}{N} \right),$$
$$\gamma_i = \left(1 - b^i\right)^k, \quad i = 1, \ldots, N, \tag{19}$$

where $N$ is the number of discount scales and $k$ is the hyperbolic parameter. This schedule will better approximate the hyperbolic integral representation. We use this method to get the fusion $Q$. The key hyper-parameters specific to our implementation are summarized in Table 11.

*Table 11.* Additional Hyper-parameters for Our Multi-timescale Mechanism implementation.

| Gamma-net implementation Hyper-parameters | |
|---|---|
| $\gamma_{\min}$ | 0 |
| $\gamma_{\max}$ | 0.99 |
| $\tau_{\min}$ | 0 |
| $\tau_{\max}$ | 100 |
| $N$ | 8 |
| **Laplace implementation Hyper-parameters** | |
| order $k$ | 2 |
| $C_k$ | 0.5 |
| **Hyperbolic implementation Hyper-parameters** | |
| $\gamma_{\max}$ | 0.99 |
| $N$ | 10 |
| hyperbolic $k$ | 0.1 |

# B. Detailed Learning Curves

We provide in this section the full training curves of our method for the main results, including the 6 DMControl common tasks (Figure 8), the 5 DMControl hard tasks (Figure 9), and the 4 different weather conditions in the CARLA benchmark (Figure 10). For reference, we also plot the training curves of DrQ-v2 under the same training setup.

# C. Extended Empirical Analysis

This section provides additional experimental data to further validate the efficiency, scalability, and structural necessity of the proposed framework, addressing the detailed computational trade-offs and component contributions.

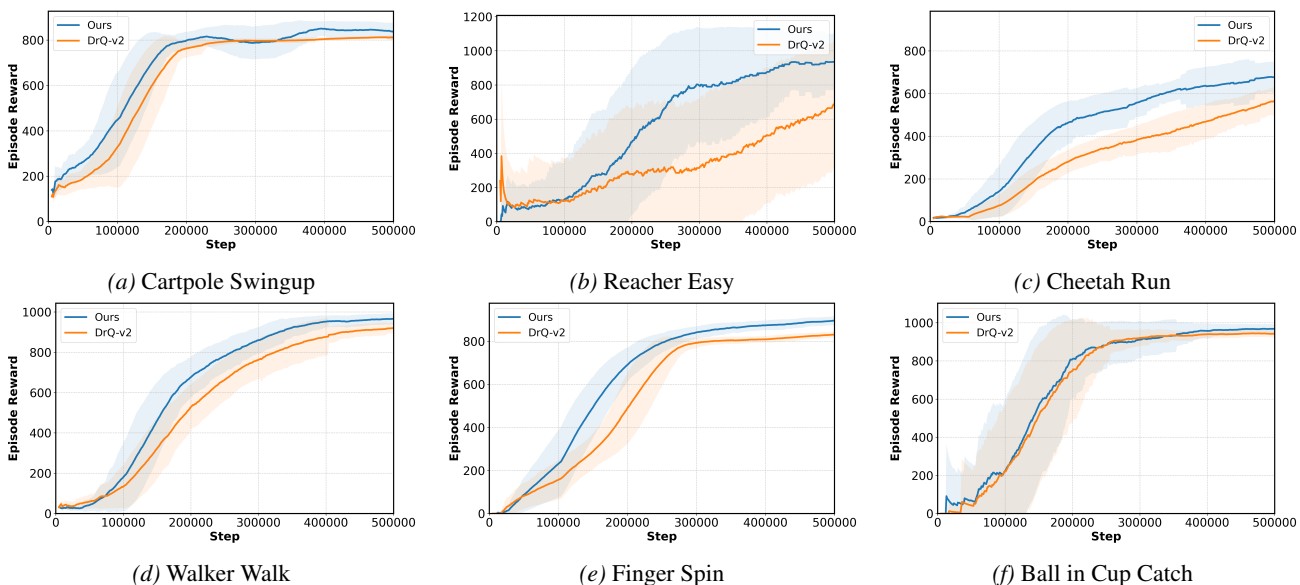

*Figure 8.* Learning curves comparison of our method and DrQ-v2 across 6 DMControl common tasks under 5 random seeds.

## C.1. Wall-clock Training Efficiency

To provide a granular view of the computational costs, we tracked evaluation scores against actual wall-clock training time on a single NVIDIA RTX 4090 GPU. The results for the *Quadruped Walk* and *Walker Run* tasks are detailed in Table 12.

Although our multi-timescale architecture introduces auxiliary objectives, the framework's superior sample efficiency effectively offsets the per-step computational overhead. Both the *Independent* and *Shared Trunk* implementations reach high-performance thresholds significantly faster than the baselines.

*Table 12.* Wall-clock training efficiency. Evaluation scores are recorded at 30-minute intervals on a single RTX 4090 GPU. The symbol "-" indicates that the training has reached its 1M step limit.

| Quadruped Walk | 0.5h | 1.0h | 1.5h | 2.0h | 2.5h | 3.0h | 3.5h | 4.0h |
|---|---|---|---|---|---|---|---|---|
| DrQ-v2 (Base) | 147 | 290 | 364 | 715 | 732 | - | - | - |
| DrQ-v2 (Widened) | 124 | 269 | 321 | 413 | 673 | 702 | 710 | 737 |
| Ours (Shared Trunk) | **249** | **345** | **510** | **762** | **797** | **812** | **830** | - |
| Ours (Independent) | 191 | 327 | 413 | 737 | 756 | 783 | 804 | 840 |

| Walker Run | 0.5h | 1.0h | 1.5h | 2.0h | 2.5h | 3.0h | 3.5h | 4.0h |
|---|---|---|---|---|---|---|---|---|
| DrQ-v2 (Base) | 163 | 236 | 415 | 505 | 538 | - | - | - |
| DrQ-v2 (Widened) | 161 | 195 | 301 | 392 | 472 | 501 | 525 | 558 |
| Ours (Shared Trunk) | **204** | **333** | **463** | **579** | **642** | **699** | **721** | - |
| Ours (Independent) | 188 | 275 | 422 | 523 | 588 | 644 | 677 | 720 |

## C.2. Scalability in High-Dimensional Perception

To evaluate the framework's scalability, we conducted experiments in the CARLA environment with a 5-camera setup, increasing the visual input dimensionality. In this high-dimensional setting, the *Widened DrQ-v2* baseline was expanded to a hidden dimension of 6144 to match the 382M parameter budget of our *Independent* implementation.

As shown in Table 13, our algorithm effectively handles complex visual inputs. Notably, the *Shared Trunk* implementation drastically optimizes efficiency, maintaining superior performance with only 43.94M parameters and 2.68G FLOPs. This confirms that the proposed consistency constraints and multi-scale architecture scale effectively to large-scale perception

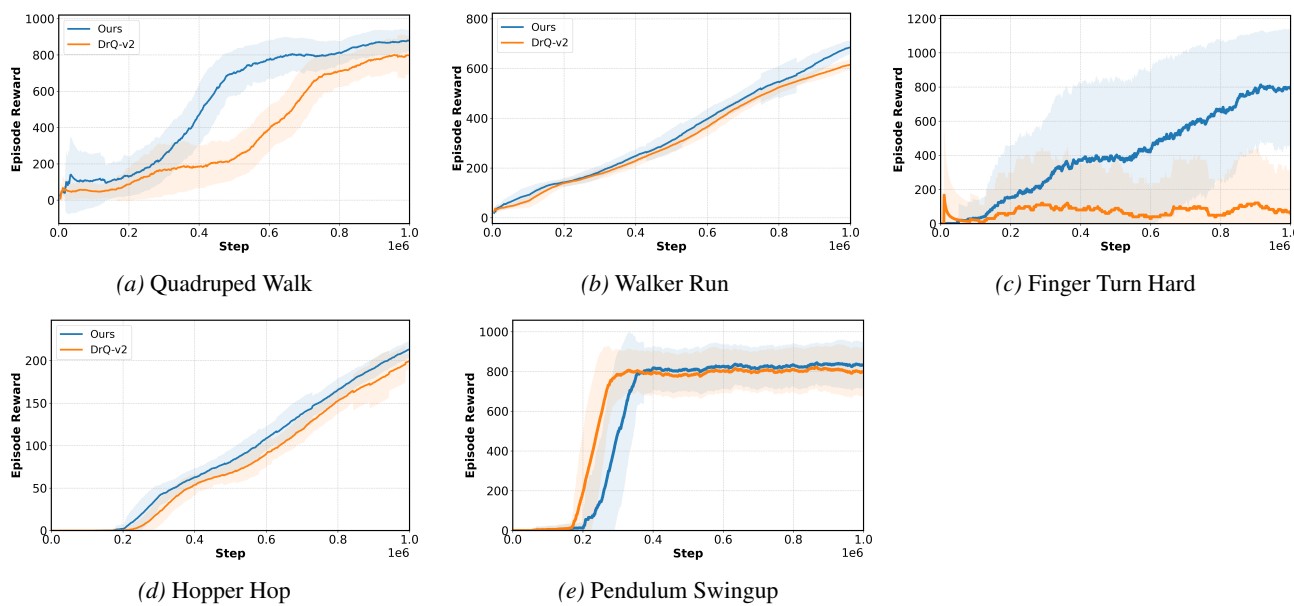

*(a)* Quadruped Walk          *(b)* Walker Run          *(c)* Finger Turn Hard

*(d)* Hopper Hop          *(e)* Pendulum Swingup

*Figure 9.* Learning curves comparison of our method and DrQ-v2 across 5 DMControl hard tasks under 5 random seeds.

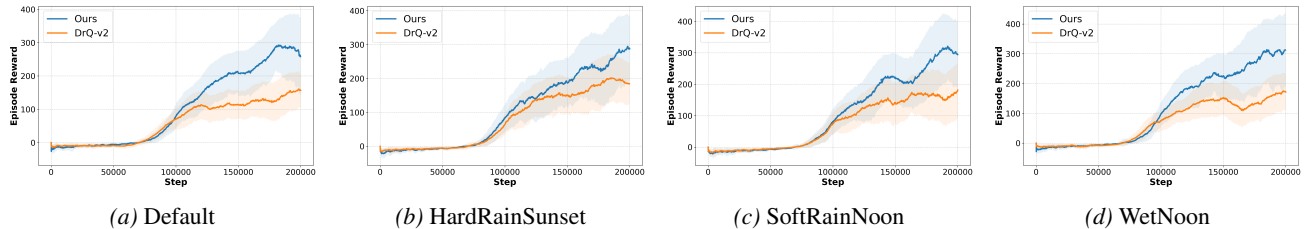

*(a)* Default          *(b)* HardRainSunset          *(c)* SoftRainNoon          *(d)* WetNoon

*Figure 10.* Learning curves comparison of our method and DrQ-v2 across 4 CARLA weathers under 5 random seeds.

tasks without necessitating prohibitive computational costs.

*Table 13.* CARLA 5-camera scalability experiment (200k steps). **Train FLOPs** denotes the total theoretical compute per RL update step. Performance is reported across different weather conditions.

| Method | Default | HardRainNoon | Params | Train FLOPs |
|---|---|---|---|---|
| DrQ-v2 (Base) | $253.6 \pm 36$ | $264.1 \pm 10$ | 30.01M | 2.47G |
| DrQ-v2 (Widened) | $261.2 \pm 29$ | $269.2 \pm 7$ | 381.6M | 6.22G |
| Ours (Shared Trunk) | $363.1 \pm 16$ | $360.1 \pm 15$ | **43.94M** | **2.68G** |
| Ours (Independent) | $\mathbf{369.2 \pm 18}$ | $\mathbf{364.3 \pm 20}$ | 381.9M | 5.05G |

## C.3. Explicit Staged Decomposition

Following the component analysis in Section 5, we provide an explicit staged decomposition by progressively adding components to the baseline. As validated in Table 14, each module contributes uniquely: (1) the multi-critic structure provides the temporal perspective; (2) reconstruction objectives ground these representations via consistency regularization; and (3) cross-attention provides context-dependent weighting. The synergy of these components drives the significant improvements in both sample efficiency and final performance.

*Table 14.* Staged decomposition analysis. Results are mean $\pm$ std at 500k and 1M environment steps.

| Method Progression | Quadruped Walk | | Walker Run | |
|---|---|---|---|---|
| | 500k Steps | 1M Steps | 500k Steps | 1M Steps |
| DrQ-v2 (Base) | $343 \pm 106$ | $732 \pm 91$ | $436 \pm 143$ | $538 \pm 115$ |
| + Multi-Critic | $366 \pm 45$ | $767 \pm 120$ | $447 \pm 15$ | $668 \pm 31$ |
| + Recon & Reward Loss | $373 \pm 88$ | $794 \pm 60$ | $469 \pm 21$ | $694 \pm 12$ |
| + Attention Fusion (Full) | $\mathbf{734 \pm 31}$ | $\mathbf{846 \pm 25}$ | $\mathbf{509 \pm 43}$ | $\mathbf{721 \pm 9}$ |

## C.4. Robustness to Sparse and Noisy Rewards

Real-world continuous control tasks often suffer from imperfect reward signals, such as extreme sparsity and sensory noise. To further demonstrate the robustness of our multi-timescale framework, we evaluated its performance under these challenging conditions.

**Resilience to Sparse Rewards.** We evaluated sparsity on *Pendulum Swingup* and *Reacher Hard*, selected for their well-defined binary states. The step-wise reward is strictly binarized: 1 if $r > 0.99$, else 0. While extreme sparsity inherently degrades absolute performance across all algorithms, our method demonstrates significant resilience (Table 15). The reward decoder seamlessly fits zero-reward sequences, allowing the framework to explicitly propagate rare success signals backward in time.

*Table 15.* Performance comparison under strictly sparse reward settings. Results are reported as mean $\pm$ std at 500k and 1M environment steps.

| Method | Pendulum Swingup | | Reacher Hard | |
|---|---|---|---|---|
| | 500k Steps | 1M Steps | 500k Steps | 1M Steps |
| DrQ-v2 (Base) | $210 \pm 101$ | $301 \pm 113$ | $89 \pm 7$ | $344 \pm 65$ |
| Ours | $\mathbf{274 \pm 27}$ | $\mathbf{420 \pm 37}$ | $\mathbf{349 \pm 88}$ | $\mathbf{485 \pm 12}$ |

**Mitigation of Noisy Rewards.** To evaluate robustness against environmental noise, we injected step-wise zero-mean Gaussian noise $\mathcal{N}(0, 0.1^2)$ into the true reward signals before storing them in the replay buffer. Unlike standard RL methods that easily overfit and suffer severe performance drops (Table 16), our framework effectively mitigates variance. The bidirectional consistency losses act as a strong regularizer, anchoring the multi-timescale representations and preventing individual Q-value estimates from succumbing to random fluctuations.

*Table 16.* Performance comparison under noisy reward settings. Results are reported as mean $\pm$ std at 500k and 1M environment steps.

| Method | Quadruped Walk | | Walker Run | |
|---|---|---|---|---|
| | 500k Steps | 1M Steps | 500k Steps | 1M Steps |
| DrQ-v2 (Base) | $313 \pm 147$ | $621 \pm 117$ | $368 \pm 12$ | $521 \pm 21$ |
| Ours | $\mathbf{491 \pm 162}$ | $\mathbf{781 \pm 70}$ | $\mathbf{429 \pm 11}$ | $\mathbf{652 \pm 14}$ |

