# OpenReview forum: "Multi-timescale Reinforcement Learning by Value Reconstruction"
_ICML.cc/2026/Conference — ICML 2026 regular_

### Official Review · Reviewer_T9pG · 2026-03-06

**Soundness:** 3
**Presentation:** 2
**Significance:** 3
**Originality:** 3
**Overall Recommendation:** 4
**Confidence:** 3

**Summary:**

The paper provides a method to incorporate multi-timescale critic in both on-policy and off-policy algorithms. The method is inspired from the findings in the field of neuroscience that points towards the presence of multi-timescale reasoning as part of cognitive process. The paper first proposes how to obtain value function for different values of discount factor and then provides a way to use the multi-timescale value function for policy update. The efficacy of the proposed method is shown in the empirical experiments where the proposed method is shown to outperform baseline methods. Further, the ablation studies further substantiates the role of multi-timescale value function.

**Compliance With Llm Reviewing Policy:**

Affirmed.

**Final Justification:**

I am satisfied with the responses of the author, and I will maintain my score.

**Key Questions For Authors:**

1. In Eq. 1, did you mean to write $\mathcal{V}\_{\phi}$  instead of $\mathcal{V} \phi$ ? Again in the last paragraph of section 3.1, is it $Q_{\phi}$ or $Q\phi$?
2. Is reward decoding horizon the episode length for the tasks ?
3. How is Table 5 strengthening the claim of the paper if the reward based performance doesn't show any variation for different values of N and T?

**Limitations:**

Yes

**Strengths And Weaknesses:**

Strength
1. The proposed method is grounded in findings in neuroscience.
2. The proposed method works for both on-policy and off-policy algorithms.
3. The experimental section is exhaustive and ablation study highlight the importance of the different components of the proposed method.

Weaknesses
1. In section 3.1 please add space after full stop.
2. In the last paragraph of section 3.1, it mentions Q-value gradient but in the parenthesis it says $Q\phi$. Please clarify this. Further, presence of $Q_{\phi}(s, \pi_{\theta}(s))$ is a characteristic of deterministic policy not the off-policy nature.
3. Certain notations are not clear. How do you obtain $\hat{r}_0$ in Eq. 9 ? Further, what does $t$ denote in Eq. 10 ?
4. It might be better to use same symbol for $\mathcal{E}_{fused}$ n Eq. 14 and Algorithm 1 line 22,23.
5. Can you please specify how the critic V(s_t) is being obtained ? On page 4, it seems it is obtained using $V(s_t) \approx LR_{0:T-1}$. From Algorithm 1, it appear the paper is using parameterized critic and updating them using $\mathcal{L}_{critic}$.
6. While defining loss functions please specify the parameters being used for the loss function. For example, $\mathcal{recon}$ doesn't provide information about parameters involved in the loss calculation.
7. In the table 5, horizontal axis denotes reward length. However, in the text the reward length is described using the term reward decoding horizon (T). If these two are the same quantity it would be better to use a consistent term for the same. Please introduce the symbol T in Table 5 as well.

Most of my concerns are related to the presentation.

---

> ### Author Rebuttal · Authors · 2026-03-30
>
> Dear Reviewer T9pG,
>
> We deeply appreciate your meticulous review. Your attention to our theoretical presentation and mathematical notations is invaluable for elevating the rigor of our paper. We sincerely apologize for the typographical omissions and will implement all your suggested corrections in the camera-ready version.
>
> 1. Typographical and Notational Corrections (Addressing W1, W4, W6, Q1)
>
> We sincerely thank the reviewer for the constructive suggestion.
>
> To clarify the optimization targets, we will update the loss functions by explicitly appending subscripts corresponding to our network components (Shared Encoder $E$, Multi-Critics $\theta$, Reward Decoder $f_\psi$, Actor $\pi_\phi$, and Attention $g_\omega$) as follows:
>
> $$\mathcal{L}\_{total}(\theta, f\_{\psi}, E) = \mathcal{L}\_{critic}(\theta, E) + \mathcal{L}\_{reward}(\theta, f\_{\psi}, E) + \mathcal{L}\_{recon}(\theta, f\_{\psi}, E)$$
>
> $$\mathcal{L}\_{actor}(\pi\_{\phi}, g\_{\omega}) = - \mathbb{E}[ \mathcal{E}\_{fused}(s, a; \theta\_{fixed}, g\_{\omega}) ]$$
>
> Additionally, we will correct the spacing in Section 3.1, fix the typo from $V\phi$ to $V_\phi$ in Eq. 1, and unify the notation for $E_{fused}$ across the manuscript, including Eq. 14 and Algorithm 1.
>
> 2. Clarification on Policy Gradients (Addressing W2 & Q1)
>
> We sincerely thank the reviewer for pointing out this imprecision. You are absolutely right that $\pi_{\theta}(s)$ is a characteristic of deterministic policies, not a universal trait of off-policy methods. And we apologize for mistakenly writing "Q-value gradients" instead of "Q-value objectives,"
>
> We will correct the final paragraph of Section 3.1 in the revision to explicitly define the optimization metric $\mathcal{M}$ using concrete examples:
>
> "Here, $\mathcal{M}$ represents the optimization metric guiding the policy update. For off-policy algorithms, $\mathcal{M}$ maximizes Q-value objectives (e.g., $Q_{\phi}(s, \pi_{\theta}(s))$ for DrQ-v2, or entropy-augmented Q-values for SAC), while on-policy algorithms like PPO utilize advantage estimates, such as Generalized Advantage Estimation (GAE) (Schulman et al., 2015)."
>
> 3. Clarification on Variables and Critic Outputs (Addressing W3 & W5)
>
> $V(s\_t)$ and $V \approx LR$: $V(s\_t)$ is directly computed via the forward pass of the parameterized multi-critic network (Algorithm 1) and updated using TD error. The equation $V \approx LR$ is not a forward computation step for $V(s\_t)$; rather, it is a theoretical consistency constraint enforced as an auxiliary regularization loss, $\mathcal{L}\_{recon}$, during training.
>
> How $\hat{r}\_0$ is obtained: $\hat{r}\_0$ is the first element (index 0) of the reconstructed reward sequence $\hat{R}$. This sequence is directly obtained from the output of the neural reward decoder, formulated as $\hat{R} = f\_{\psi}(V(s\_t))$.
>
> What $t$ denotes: The variable $t$ denotes the current environment timestep corresponding to the state $s_t$ sampled from the buffer. Consequently, $r_t$ in Eq. 9  is the immediate ground-truth reward collected at the current step $t$, while $r_{t+k}$ in Eq. 10  refers to the ground-truth reward collected $k$ steps into the future.
>
> 4. Decoding Horizon (T) and the Significance of Table 5 (Addressing W7, Q2, Q3)
>
> Decoding Horizon ($T$): $T$ (e.g., $T=15$) is a short, local sliding lookahead window, strictly not the full episode length. We will unify the terminology to "Decoding Horizon ($T$)" in Table 5.
>
> Significance of Multiple Timescales (Addressing Q3): You raised a critical question: if performance shows little variation, are the extra timescales redundant? We acknowledge that the original Table 5 used heuristically chosen $\gamma$ lists, which inadvertently masked the true necessity of scaling $N$. To replace manual ablation, we tested random gamma robustness. Across 5 trials (T=15), we generated lists with 0.95 and 7 random values in (0, 0.95). We randomly dropped two non-0.95 values iteratively to evaluate $N \in \{6, 4, 2\}$. The results below clarify our core claims:
>
> Necessity of Multi-scale: Any multi-scale configuration ($N \ge 2$) significantly outperforms the single-scale baseline (DrQ-v2), proving that multi-timescale reasoning is fundamentally beneficial.
>
> Capacity vs. Robustness: Performance improves sharply from $N=2$ to $N=4$. The performance plateau observed at $N \ge 4$ does not imply redundancy; rather, it demonstrates hyperparameter robustness. It proves that once sufficient temporal resolution is reached, the framework stabilizes—it neither degrades with extra $\gamma$ values nor requires meticulous manual tuning.
>
> | Gamma Selection | quadruped-walk (500k / 1M Steps) | walker-run (500k / 1M Steps) |
> | :--- | :--- | :--- |
> | drqv2 | 343 ± 106 / 732 ± 91 | 436 ± 143 / 538 ± 115 |
> | 2gamma | 420 ± 70 / 797 ± 34 | 442 ± 21 / 649 ± 14 |
> | 4gamma | 726 ± 40 / 830 ± 22 | 499 ± 45 / 717 ± 23 |
> | 6gamma | 720 ± 31 / 827 ± 53 | 498 ± 15 / 725 ± 5 |
> | 8gamma | 724 ± 37 / 831 ± 15 | 496 ± 27 / 726 ± 12 |

---

> > ### Author Rebuttal · Reviewer_T9pG · 2026-04-03
> >
> > Thank you for addressing my concerns. I have the following questions:
> > 1. Please improve the notation of the loss in Eq. 9 and Eq. 10. From the current notation, it is difficult to understand the role of timestep $t$.
> > 2. It will be good to include a small remark regarding the distinction between the Decoding horizon and the Episode Length in the paper.
> > 3. Thank you for showing the comparison of N=2 vs N=4. It will be better to show this comparison in the paper to highlight the improvement in performance.

---

> > > ### Author Response · Authors · 2026-04-04
> > >
> > > Dear Reviewer T9pG,
> > >
> > > Thank you for your constructive follow-up suggestions. We will gladly implement these refinements in the revised manuscript to improve the clarity of our paper.
> > >
> > > 1. Role of timestep $t$ in Eq. 9 and Eq. 10:
> > > Thank you for the suggestion. We will clarify that $t$ denotes the exact environment timestep when the current state $s_t$ is sampled, serving as the starting reference point. Specifically, it maps the decoder's relative forward step $k$ to the actual environment step $t+k$, ensuring each predicted reward $\hat{r}\_{k}$ is aligned with the correct ground-truth reward $r_{t+k}$. We will add this explanation immediately after Eq. 9 and Eq. 10.
> > >
> > > 2. Distinction between Decoding Horizon and Episode Length:
> > > Thank you for pointing this out. We will add a remark in Section 3.2 to explicitly distinguish the two concepts: the Decoding Horizon $T$ (e.g., $T=15$) is a short, local sliding lookahead window strictly used for auxiliary reward reconstruction. It is mathematically decoupled from, and significantly shorter than, the environment's full Episode Length (which is typically set to 1000 in benchmarks like DMControl and CARLA).
> > >
> > > 3. Comparison of $N=2$ vs $N=4$:
> > > Thank you for the suggestion. We will incorporate the random $\gamma$ ablation results—explicitly demonstrating the performance improvement from $N=2$ to $N=4$—directly into the main text of the revised manuscript.
> > >
> > > Thank you again for your invaluable guidance!

---

### Official Review · Reviewer_PywD · 2026-03-08

**Soundness:** 3
**Presentation:** 3
**Significance:** 3
**Originality:** 3
**Overall Recommendation:** 4
**Confidence:** 3

**Summary:**

This paper proposes a multi-timescale RL framework that augments a base actor-critic method with multiple critics defined at different discount factors, a neural reward decoder, and consistency objectives based on both value reconstruction and reward reconstruction. The reconstructed multi-timescale representations are fused through a cross-attention module and then used for policy learning. The method is evaluated on DMControl and CARLA, and is integrated with several standard RL backbones including SAC, PPO, and DrQ-v2. Empirically, the paper reports gains over these baselines while maintaining similar inference-time latency.

**Compliance With Llm Reviewing Policy:**

Affirmed.

**Final Justification:**

The rebuttal addressed my concerns

**Key Questions For Authors:**

1. Can the authors include parameter-matched or capacity-controlled baselines, for example by widening the backbone critics of the underlying RL algorithms to a similar parameter budget? This would help clarify whether the observed improvements truly come from the multi-timescale design.

2. Can the authors compare cross-attention against simpler adaptive fusion schemes, such as gating or learned weighted averaging? This would better justify the architectural choice and clarify whether attention is necessary.

3. Can the ablation study be expanded into a more explicit decomposition, e.g., base method -> + multi-critic -> + reconstruction losses -> + learned fusion -> full model? This would make it easier to attribute gains to specific components.

**Limitations:**

See weakness

**Strengths And Weaknesses:**

Strengths:

1. The paper addresses a meaningful design question in deep RL: how to exploit predictive structure at multiple temporal scales without committing to a single discount factor. The overall method is clearly organized around this motivation, and the proposed reconstruction objectives provide a concrete way to tie together information across horizons.

2. The method is more than a simple multi-head critic extension. The combination of multi-timescale critics, reward/value reconstruction, and a learned fusion module gives the approach a coherent algorithmic identity, rather than presenting the gains as coming from a loosely assembled set of heuristics.

3. The empirical evaluation is reasonably broad in the sense that the approach is tested across multiple domains and plugged into several common RL algorithms. This improves confidence that the idea is not narrowly tied to a single environment or implementation stack.

4. The paper makes a useful practical point that inference latency remains close to the underlying baseline despite the richer training objective. If the performance gains hold under stronger controls, this would make the method appealing for settings where training complexity is more tolerable than deployment overhead.

Weaknesses:

1. The main empirical comparisons do not adequately control for model capacity. The paper increases parameters from roughly 7.3M to 69.5M, which is large enough that capacity becomes a serious confounder when interpreting the gains. Exact parameter matching is not always necessary in RL papers, but a nearly 10x increase makes it difficult to tell how much of the improvement comes from the proposed multi-timescale structure versus simply using a much larger function approximator.

2. The evidence supporting the specific fusion design is still limited. The paper compares cross-attention mainly against a simple Q-average baseline, which is not sufficient to establish that attention is the right performance-complexity trade-off. Stronger evidence would require comparison to lighter adaptive fusion mechanisms, such as learned gating or MLP-based weighting, to show that the gains are due to the representational role of attention rather than just additional flexibility.

3. The ablations are helpful but still not fully diagnostic about where the gains come from. Because the full method combines several coupled components, the current experiments do not yet cleanly separate the contributions of (i) multi-critic structure, (ii) reconstruction losses, and (iii) learned fusion. A more explicit staged decomposition would make the causal story behind the reported improvements much more convincing.

4. The paper argues that inference cost stays close to the baseline, but the training-side overhead is less systematically characterized. Since the method introduces many more parameters and extra objectives, it would be helpful to understand the trade-off under matched training budget, wall-clock time, or compute. This matters because the practical value of the method depends not only on deployment efficiency but also on whether the gains remain favorable under realistic training-cost constraints.

---

> ### Author Rebuttal · Authors · 2026-03-30
>
> Dear Reviewer PywD, thank you for recognizing our framework's value. We have conducted extensive experiments to address your insightful queries.
>
> 1.Model Capacity vs. Algorithmic Design (Addressing W1 & W4 & Q1)
>
> Widened Baseline (69M). We widened DrQ-v2 by increasing the critic's hidden dimension from 1024 to 4608 to match our Full model's budget, which yields only marginal gains.
>
> Reduced Version (18M). To address our full model's parameter overhead, we optimized the architecture by: (1) sharing the visual trunk across all critics; (2) sharing the first two layers of the Q1/Q2 MLPs as a unified Q-trunk (keeping final linear layers separate); and (3) utilizing this shared trunk as the attention query. This slashes our parameters while preserving SOTA performance within standard variance.
>
> || Quadruped Walk (500k / 1M Steps) | Walker Run (500k / 1M Steps) | Params |
> | :--- | :---: | :---: | :---: |
> | DrQ-v2 (Base) | 343 ± 106 / 732 ± 91 | 436 ± 143 / 538 ± 115 | 7M |
> | Widened DrQ-v2 | 387 ± 71 / 760 ± 82 | 477 ± 62 / 558 ± 67 | 69M |
> | Gamma-Nets | 152 ± 63 / 364 ± 73 | 234 ± 30 / 431 ± 40 | 7M |
> | Hyperbolic | 175 ± 87 / 744 ± 130 | 392 ± 15 / 687 ± 25 | 45M |
> | Laplace | 355 ± 21 / 788 ± 144 | 483 ± 26 / 681 ± 53 | 61M |
> | REDQ | 308 ± 101 / 756 ± 54 | 280 ± 15 / 582 ± 37 | 11M |
> | Ours (Reduced) | 726 ± 46 / 830 ± 54 | 496 ± 24 / 722 ± 10 | **18M** |
> | Ours (Full) | **734 ± 31** / **846 ± 25** | **509 ± 43** / **721 ± 9** | 69M |
>
> Other multi-critic methods (e.g., Hyperbolic, Laplace) inherently introduce large parameter counts but fail to yield commensurate SOTA performance.
>
> Wall-clock Time & Training Efficiency. We also tracked evaluation scores against wall-clock training time on a single RTX 4090 ("-" indicates completion of 1M training steps) . Thus, our extra objectives provide a highly favorable performance-compute trade-off under realistic training-cost constraints.
>
> | Quadruped Walk| 0.5h | 1.0h | 1.5h | 2.0h | 2.5h | 3.0h | 3.5h | 4.0h |
> | :--- | :---: | :---: | :---: | :---: | :---: | :---: | :---: | :---: |
> | DrQ-v2 (Base) | 147 | 290 | 364 | 715 | 732 | - | - | - |
> | DrQ-v2 (Widened) | 124 | 269 | 321 | 413 | 673 | 702 | 710 | 737 |
> | Ours (Full) | 191 | 327 | 413 | 737 | 756 | 783 | 804 | 840 |
> | Ours (Reduced) | 249 | 345 | 510 | 762 | 797 | 812 | 830 | - |
>
> | Walker Run| 0.5h | 1.0h | 1.5h | 2.0h | 2.5h | 3.0h | 3.5h | 4.0h |
> | :--- | :---: | :---: | :---: | :---: | :---: | :---: | :---: | :---: |
> | DrQ-v2 (Base) | 163 | 236 | 415 | 505 | 538 | - | - | - |
> | DrQ-v2 (Widened) | 161 | 195 | 301 | 392 | 472 | 501 | 525 | 558 |
> | Ours (Full) | 188 | 275 | 422 | 523 | 588 | 644 | 677 | 720 |
> | Ours (Reduced) | 204 | 333 | 463 | 579 | 642 | 699 | 721 | - |
>
> 2. Justifying the Fusion Design (Addressing W2 & Q2)
>
> We evaluated a 2-layer MLP baseline processing concatenated states and multi-scale Q-values into softmax weights. While better than averaging, it falls short of cross-attention. Unlike MLPs that rigidly project flat vectors, cross-attention uses the state as a query to dynamically attend to each critic's distinct features, matching state semantics with specific timescale properties.
>
> || Quadruped Walk (500k / 1M Steps) | Walker Run (500k / 1M Steps) |
> | :--- | :---: | :---: |
> | DrQ-v2 (Base) | 343 ± 106 / 732 ± 91 | 436 ± 143 / 538 ± 115 |
> | Ours w/ MLP | 656 ± 104 / 752 ± 38 | 456 ± 21 / 697 ± 44 |
> | Ours (Cross-Attention) | 734 ± 31 / 846 ± 25 | 509 ± 43 / 721 ± 9 |
>
> While attention-based multi-timescale fusion is our novel contribution, it is well supported by existing RL research. In MARL, QATTEN and NA2Q show that standard MLP mixers have representational bottlenecks in aggregating Q-values, while attention allows flexible, context-dependent weighting. This strongly justifies our cross-attention design over rigid MLPs.
>
> [1]Qatten: A general framework for cooperative multiagent reinforcement learning
>
> [2]N $\text {A}^{\text {2}} $ Q: Neural Attention Additive Model for Interpretable Multi-Agent Q-Learning
>
> 3. Explicit Staged Decomposition (Addressing W3 & Q3)
>
> Following your suggestion, we expanded the ablation into a staged decomposition. Results cleanly validate that each component plays an indispensable and distinct role: (1) The multi-critic structure provides a comprehensive multi-timescale perspective; (2) The reconstruction objectives serve as a crucial regularization mechanism to ground these representations; and (3) The attention module provides the dynamic focusing capability. The synergistic effect of all three components drives the significant gains in both final performance and early sample efficiency.
>
> || Quadruped Walk (500k / 1M Steps) | Walker Run (500k / 1M Steps) |
> | :--- | :---: | :---: |
> | DrQ-v2 (Base) | 343 ± 106 / 732 ± 91 | 436 ± 143 / 538 ± 115 |
> | + Multi-Critic | 366 ± 45 / 767±120 | 447 ± 15 / 668±31 |
> | + Recon & Reward Loss | 373 ± 88 / 794 ± 60 | 469 ± 21 / 694 ± 12 |
> | + Attention Fusion (Full) | 734 ± 31 / 846 ± 25 | 509 ± 43 / 721 ± 9 |

---

> > ### Author Rebuttal · Reviewer_PywD · 2026-04-03
> >
> > The rebuttal addresses most of my concerns and substantially strengthens the paper. In particular, the added parameter-controlled baselines, reduced-capacity variant, and wall-clock comparisons significantly reduce the main empirical confounds I raised, while the new fusion comparison and staged ablation make the role of each component much clearer. Although additional fusion baselines could still further strengthen the work, I believe the rebuttal has resolved the main issues to a satisfactory degree. I am therefore raising my score from 3 to 4.

---

> > > ### Author Response · Authors · 2026-04-03
> > >
> > > Dear Reviewer PywD,
> > >
> > > Thank you very much for your time and for raising the score. Your constructive feedback has been incredibly meaningful in strengthening our work. We will ensure that all these new contents and discussions are included in the revised manuscript.
> > >
> > > Thank you again for your support!

---

### Official Review · Reviewer_XTLF · 2026-03-12

**Soundness:** 3
**Presentation:** 2
**Significance:** 3
**Originality:** 3
**Overall Recommendation:** 4
**Confidence:** 4

**Summary:**

This paper proposes a multi-timescale framework that introduces a Neural Reward Decoder for consistent Q-value estimation through value and reward reconstruction losses, and a cross-attention-based Q-weight predictor to adaptively fuse multi-scale Q-values for policy optimization. Experiments show this plug-and-play module significantly outperforms state-of-the-art baselines on DMControl and CARLA benchmarks, showcasing strong generalizability and consistent performance gains across both off-policy and on-policy RL algorithms.

**Compliance With Llm Reviewing Policy:**

Affirmed.

**Final Justification:**

The rebuttal and discussion have addressed most of my concerns. The proposed method makes sense and seems to be robust to hyperparameter choice. I would suggest a more in-depth discussion about the new objective function that is optimized in the proposed method, which is currently missing. For instance, if the original goal is to maximize an expected discounted return with a fixed and known discount factor, what's the impact of using the proposed method with respect to this original goal?
Overall, I'd like to keep my current score.

**Key Questions For Authors:**

1. In Section 4.2, it’s said that "This allows the model to learn a smooth manifold…", what is this manifold to be learnt?
2. How to choose the corresponding $\gamma$ list besides avoiding close spacing? Is the performance sensitive to the choice of this list?
3. How does different critic count $N$ and decoding horizon $T$ affect the complexity and inference efficiency?

**Limitations:**

As mentioned in Section 5.6, the training cost and parameter count sharply increases.
The sensitivity to the $\gamma_i$'s is not fully explored.
For off-policy methods, supervising only immediate reward might be weak, especially if $V(s_t)$ itself is biased and noisy.

**Strengths And Weaknesses:**

**Soundness**

The proposed method draws inspiration from recent work in neuroscience and is relatively well-motivated.
In particular, the use of Recon loss and Reward loss for the decoder, designed for both on-policy and off-policy methods, is theoretically sound and empirically verified.
The effectiveness of the whole architecture demonstrates strong performance gain and stability.

The proposed method seems to optimize a novel criterion in reinforcement learning. I would suggest the authors comment more on this point. In addition, they could also explain the meaning of the scores in the experimental results.

**Presentation**

Although the proposed architecture is complicated, the authors successfully developed the whole picture logically step by step, with comparison to related work clearly addressed. The various components used are supported with ablation studies.

Some notations are not explained (e.g., \hat r_k) and there are a few number of typos, e.g.
- In equation (1) it should be V_{\phi} instead of V\phi. Actually, many terms should be in subscript.
- \Epsilon_{fused} vs \Epsilon_{agg}

**Significance**

The paper tackles the following issues: (1) traditional reinforcement learning agents, relying on a single discount factor, struggle to balance short-term objectives and long-term planning, and (2) existing multi-timescale methods lack explicit consistency enforcement or adaptive value fusion. This study addresses these issues with novel value reconstruction and fusion approach, which is applicable to both on-policy and off-policy methods.

**Originality**

The paper uses a learnable decoder for approximating inverse mappings, with novel extensions by introducing the reward-reconstruction loss and the multi-Q fusion by cross-attention.
By these original designs, the paper distinguishes from alternatives (Gamma-Nets, Laplace, Hyperbolic) and outperforms them.

---

> ### Author Rebuttal · Authors · 2026-03-30
>
> Dear Reviewer XTLF,
>
> We thank you for recognizing our framework's value. We address your queries below.
>
> 1.Clarifications on $\\hat{r}\_{k}$ and Notations
>
> We sincerely apologize for the typographical omissions. Regarding $\\hat{r}\_{k}$ in Eq. (10), it denotes the k-th element of the reconstructed reward sequence $\\hat{R}$ generated by the Neural Reward Decoder. It represents the predicted reward for the k-th forward step within the decoding horizon T. We will explicitly add this definition, correct the subscript in Eq. (1) from $V\\phi$ to $\\mathcal{V}\_{\\phi}$, and unify the notation to consistently use $\\mathcal{E}\_{fused}$ instead of $\\mathcal{E}\_{agg}$ in the camera-ready version.
>
> 2. Clarification on the "Smooth Manifold" (Addressing Q1)
>
> Unlike direct algebraic inversion, which amplifies Q-value errors into chaotic fluctuations, our neural decoder restricts reconstructions to physically plausible sequences. Accordingly, we will replace the term "manifold" with "a space of valid reward sequences" in the revision.
>
> 3. Sensitivity to gamma Choice (Addressing Q2)
>
> Performance is highly robust to exact values given sufficient temporal resolution ($N \ge 4$). To systematically evaluate this, we tested random gamma robustness. Across 5 trials (T=15), we generated 8-scale lists comprising a fixed 0.95 and 7 random values in (0, 0.95). We iteratively dropped two non-0.95 values at random to evaluate $N \in \{6, 4, 2\}$. As shown in the Section 5 table, meticulous manual tuning is largely unnecessary. Crucially, integrating multiple gamma scales consistently outperforms single-scale baselines, with final rewards steadily increasing as N grows before reliably plateauing at $N \ge 4$ (with fluctuations strictly within standard deviation bounds). This proves the framework is highly robust to exact gamma choices and list sizes.
>
> 4. Clarification on Off-Policy Reward Supervision (Addressing Limitations)
>
> In off-policy learning, supervising $\hat{R}$ entirely with replay buffer data introduces severe bias due to distributional shift between historical behavioral policies and the updating target policy. Thus, only the immediate reward $r_t$ is unbiased. Nevertheless, our Value Reconstruction Loss ($\mathcal{L}_{recon}$) enforces strict cycle-consistency across multi-scale Q-values, robustly regularizing latent representations despite lacking full-sequence supervision.
>
> 5. Quantitative Impact of N and T on Complexity and Performance (Addressing Q3)
>
> Zero Inference Overhead: During deployment, the actor operates independently. All auxiliary modules are discarded, adding 0 parameters and 0 FLOPs to inference latency.
>
> Note: To strictly isolate architectural overhead, we report Train FLOPs as the total theoretical compute—including all corresponding forward and backward passes—for a single complete RL update step per sample
>
> Impact of T: T solely dictates the decoder's final projection. Increasing T by 1 adds <0.01M params/FLOPs (negligible overhead). Performance is stable across practical ranges ($20 \ge T \ge 10$). A too-short T degenerates to 1-step TD, while an overly long T yields diminishing returns from gamma-decay and environmental stochasticity.
>
> Impact of N on Complexity & Performance: Increasing N inherently adds computational load, with the specific growth magnitude detailed in the table below.
>
> | N | Params | FLOPs | quadruped-walk (500k/1M) | walker-run (500k/1M) |
> | :--- | :--- | :--- | :--- | :--- |
> | 1(drqv2) | 7.3M | 404.4M | 343 ± 106 / 732 ± 91 | 436 ± 143 / 538 ± 115 |
> | 2 | 52.2M | 694.4M | 420 ± 70 / 797 ± 34 | 442 ± 21 / 649 ± 14 |
> | 4 | 60.6M | 779.0M | 726 ± 40 / 830 ± 22 | 499 ± 45 / 717 ± 23 |
> | 6 | 69.0M | 863.6M | 720 ± 31 / 827 ± 53 | 498 ± 15 / 725 ± 5 |
> | 8 | 77.4M | 948.1M | 724 ± 37 / 831 ± 15 | 496 ± 27 / 726 ± 12 |
>
> Parameter Mitigation Validation: Our Reduced version (1) shares the visual trunk across critics, (2) shares the first two Q1/Q2 MLP layers, and (3) uses this shared trunk as the attention query. As confirmed below at our default setting (N=6, T=15), this slashes our parameters while preserving SOTA performance within standard variance, effectively eliminating the complexity bottleneck.
>
> || Params | FLOPs | quadruped-walk (500k / 1M) | walker-run (500k / 1M) |
> | :--- | :--- | :--- | :--- | :--- |
> | DrQ-v2 | 7M | 404.4M | 343 ± 106 / 732 ± 91 | 436 ± 143 / 538 ± 115 |
> | Widened DrQ-v2 | 69M | 974.7M | 387 ± 71 / 760 ± 82 | 477 ± 62 / 558 ± 67 |
> | Ours (Reduced)| 18M | 363.7M | 726 ± 46 / 830 ± 54 | 496 ± 24 / 722 ± 10 |
> | Ours (Full) | 69M | 863.6M | 734 ± 31 / 846 ± 25 | 509 ± 43 / 721 ± 9 |
>
> 6.Addressing Soundness
>
> Our novel optimization criterion extends standard return maximization by enforcing strict temporal consistency across multi-scale Q-values via value and reward reconstruction. Accordingly, the scores reported in our experiments strictly represent the standard, unmodified cumulative episodic rewards directly provided by the DMControl benchmark.

---

> > ### Author Rebuttal · Reviewer_XTLF · 2026-04-02
> >
> > I thank the authors for their responses, which have adequately addressed my concerns and questions.

---

> > > ### Author Response · Authors · 2026-04-03
> > >
> > > Dear Reviewer XTLF,
> > >
> > > Thank you for your time and for confirming that our rebuttal has addressed your concerns. Your constructive questions have been very helpful to us. We will ensure all the clarifications are incorporated into the revised manuscript.
> > >
> > > Thank you again for your review and support!

---

### Official Review · Reviewer_jmv2 · 2026-03-13

**Soundness:** 3
**Presentation:** 4
**Significance:** 4
**Originality:** 3
**Overall Recommendation:** 5
**Confidence:** 3

**Summary:**

The submission introduces a novel multi-timescale reinforcement learning (RL) framework that addresses the limitations of using a fixed single discount factor. The authors outline the key problem that a single timescale fails to balance immediate rewards with long-term strategic planning in complex decision-making tasks. Inspired by recent neuroscientific findings, the paper proposes a multi-critic architecture where each component corresponds to a different discount factor. The framework enhances value estimation through a Neural Reward Decoder that enforces consistency via reward reconstruction losses and utilizes a Cross-Attention-based Q-weight predictor to adaptively fuse multi-scale values based on the environmental context. Empirical evaluations on DMControl and CARLA demonstrate significant performance gains across various RL paradigms, including SAC, DrQ-v2, and PPO.

**Compliance With Llm Reviewing Policy:**

Affirmed.

**Final Justification:**

The rebuttal and follow-up responses provide substantial additional analysis addressing my concerns on training complexity and scalability, including detailed FLOPs, wall-clock comparisons, and large-scale experiments in CARLA. These results convincingly demonstrate that the method scales effectively with manageable computational overhead.

Overall, the paper is significantly strengthened, and I revise my score from 4 to 5.

**Key Questions For Authors:**

1.	Automating Timescale Selection: Given the sensitivity analysis in Table 5, did the authors observe any scenarios where a uniform distribution of $\gamma$ performed significantly worse than a manually selected set? Is it feasible to learn the discount factors end-to-end alongside the reward decoder?

2.	Impact of Reward Sparse/Noisy Signals: Since the Neural Reward Decoder relies on reconstructing the reward sequence, how does the framework perform in environments with extremely sparse rewards? If the reward is zero for long durations, does the consistency loss ($L_{recon}$) still provide meaningful gradients for value estimation?

3.	Parameter Efficiency: Could the authors elaborate on the potential for using weight-sharing or a single hypernetwork to produce multi-scale Q-values, thereby reducing the large parameter overhead observed in the current ensemble-based implementation?

**Limitations:**

yes

**Strengths And Weaknesses:**

Strengths

1.	Robust Multi-Scale Modeling: Replaces unstable analytical methods with a learnable neural decoder to approximate the inverse Laplace transform, ensuring consistency across planning horizons.

2.	Adaptive Contextual Fusion: Uses Cross-Attention to dynamically reweight timescales based on situational semantics

Weaknesses

1.	Substantial Training Overhead: While the inference latency remains low, the reviewer thinks the authors should more critically address the significantly increased training complexity. As noted in Section 5.6, the parameter count increases nearly ten-fold (from 7.28M to 69.47M), and throughput is halved. The submission lacks a detailed discussion on the hardware requirements and the scalability of this approach for larger-scale environments where memory and computational budgets are constrained.

2.	Sensitivity to the Choice of Discount Factor Sets: The framework utilizes a predefined set of discount factors ($\gamma \in \{0.3, 0.5, 0.7, 0.8, 0.9, 0.95\}$). Although the authors provide a sensitivity analysis in Table 5, it remains unclear how to systematically determine the optimal resolution and range of $\Gamma$ for a novel task. The reviewer believes that providing a guideline or a more automated way to select these timescales would enhance the generalizability of the proposed framework.

---

> ### Author Rebuttal · Authors · 2026-03-30
>
> Thank you for recognizing our framework. We have performed targeted experiments to address your insightful questions on parameter efficiency, timescale selection, and reward distributions.
>
> 1. Parameter Efficiency & Potential for Weight-Sharing (Addressing W1 & Q3)
>
> Widened Baseline (69M). We widened DrQ-v2 by increasing the critic's hidden dimension from 1024 to 4608 to match our Full model's budget, which yields only marginal gains.
>
> Reduced Version (18M). To address our full model's parameter overhead, we optimized the architecture by: (1) sharing the visual trunk across all critics; (2) sharing the first two layers of the Q1/Q2 MLPs as a unified Q-trunk (keeping final linear layers separate); and (3) utilizing this shared trunk as the attention query. This slashes our parameters while preserving SOTA performance within standard variance.
>
> | | Quadruped Walk (500k / 1M Steps) | Walker Run (500k / 1M Steps) | Params |
> | :--- | :---: | :---: | :---: |
> | DrQ-v2 (Base) | 343 ± 106 / 732 ± 91 | 436 ± 143 / 538 ± 115 | 7M |
> | Widened DrQ-v2 | 387 ± 71 / 760 ± 82 | 477 ± 62 / 558 ± 67 | 69M |
> | Gamma-Nets | 152 ± 63 / 364 ± 73 | 234 ± 30 / 431 ± 40 | 7M |
> | Hyperbolic | 175 ± 87 / 744 ± 130 | 392 ± 15 / 687 ± 25 | 45M |
> | Laplace | 355 ± 21 / 788 ± 144 | 483 ± 26 / 681 ± 53 | 61M |
> | REDQ | 308 ± 101 / 756 ± 54 | 280 ± 15 / 582 ± 37 | 11M |
> | Ours (Reduced) | 726 ± 46 / 830 ± 54 | 496 ± 24 / 722 ± 10 | 18M |
> | Ours (Full) | 734 ± 31 / 846 ± 25 | 509 ± 43 / 721 ± 9 | 69M |
>
> 2. Automating Timescale Selection & Robustness (Addressing W2 & Q1)
>
> To address this, we implemented a variant where the gamma list (originally designed to broadly span diverse timescales without meticulous tuning) is initialized with [0.3, 0.5, 0.7, 0.8, 0.9, 0.95] as learnable parameters. As shown below, it underperforms our fixed-list approach. Dynamically shifting gammas continuously alter the L matrix and regression targets, leading to training instability. Crucially, our primary objective is optimizing the policy for the largest gamma, treating others strictly as auxiliary targets. Learnable gammas tend to degenerate by improperly catering to auxiliary objectives (e.g., trivially tuning gamma to minimize reward loss), which actively hinders the optimization of the primary target.
>
> |  | quadruped-walk (500k / 1M Steps) | walker-run (500k / 1M Steps) |
> | :--- | :--- | :--- |
> | drqv2 | 343 ± 106 / 732 ± 91 | 436 ± 143 / 538 ± 115 |
> | ours (E2E) | 538 ± 121 / 825 ± 52 | 490 ± 28 / 715 ± 19 |
> | ours | 734 ± 31 / 846 ± 25 | 509 ± 43 / 721 ± 9 |
>
> To replace manual ablation, we tested random gamma robustness. Across 5 trials (T=15), we generated lists with 0.95 and 7 random values in (0, 0.95). We randomly dropped two non-0.95 values iteratively to evaluate $N \in \{6, 4, 2\}$. As demonstrated below, increasing timescales consistently yields higher final rewards than the single-scale baseline. Performance significantly improves and then plateaus once a sufficient temporal resolution (N >= 4) is reached.
>
> |  | quadruped-walk (500k / 1M Steps) | walker-run (500k / 1M Steps) |
> | :--- | :--- | :--- |
> | drqv2 | 343 ± 106 / 732 ± 91 | 436 ± 143 / 538 ± 115 |
> | 2gamma | 420 ± 70 / 797 ± 34 | 442 ± 21 / 649 ± 14 |
> | 4gamma | 726 ± 40 / 830 ± 22 | 499 ± 45 / 717 ± 23 |
> | 6gamma | 720 ± 31 / 827 ± 53 | 498 ± 15 / 725 ± 5 |
> | 8gamma | 724 ± 37 / 831 ± 15 | 496 ± 27 / 726 ± 12 |
>
> Remarkably, the random lists achieve performance highly comparable to our empirically chosen list (with any gaps strictly within standard deviation bounds), firmly validating the framework's strong robustness. Crucially, this reinforces a key conclusion : as long as the gamma list is fixed (even randomly), it effectively circumvents the target drift and instability inherent to E2E learnable gammas. Thus, while meticulous manual tuning is fundamentally unnecessary, future work will explore task-level adaptive gamma initialization.
>
> 3. Impact of Sparse and Noisy Rewards (Addressing Q2)
>
> Sparse : We evaluated sparsity on pendulum_swingup and reacher_hard, as these goal-oriented tasks have well-defined binary states (1 if $\ge 0.99$, else 0). While extreme sparsity reduces all algorithms’ absolute performance, our framework is highly resilient. Our reward_decoder can fit zero-reward sequences without contradiction, explicitly propagating rare success signals backward in time.
>
> |  | pendulum_swingup (500k / 1M) | reacher_hard (500k / 1M) |
> | :--- | :--- | :--- |
> | drqv2 | 210 ± 101 / 301 ± 113 | 89 ± 7 / 344 ± 65 |
> | ours | 274 ± 27 / 420 ± 37 | 349 ± 88 / 485 ± 12 |
>
> Noisy : Unlike standard RL that easily overfits step-wise noise, our framework mitigates variance through our consistency losses act as a strong regularizer that prevents individual Q-values from random fluctuations.
>
> |  | quadruped-walk (500k / 1M) | walker-run (500k / 1M) |
> | :--- | :--- | :--- |
> | drqv2 | 313 ± 147 / 621 ± 117 | 368 ± 12 / 521 ± 21 |
> | ours | 491 ± 162 / 781 ± 70 | 429 ± 11 / 652 ± 14 |

---

> > ### Author Rebuttal · Reviewer_jmv2 · 2026-04-04
> >
> > We thank the authors for the detailed and well-structured responses, as well as the additional experiments. The analysis on parameter efficiency, including the reduced model and weight-sharing design, is particularly helpful, and the results on timescale selection and reward sparsity further strengthen the empirical support of the method.
> >
> > However, my concern regarding the overall training complexity and scalability remains only partially addressed. While the parameter count is reduced, a more comprehensive analysis of the computational cost, including training efficiency and scalability to larger environments, would be necessary to fully assess the practicality of the approach.
> >
> > Given these remaining concerns, I will maintain my original score.

---

> > > ### Author Response · Authors · 2026-04-07
> > >
> > > 1.Wall-clock Time & Training Efficiency.
> > >
> > > To comprehensively address your remaining concerns regarding overall training complexity, we provide a detailed computational analysis of the four architectural variants established in our previous response. We tracked evaluation scores against wall-clock training time on a single RTX 4090 ("-" indicates completion of 1M training steps).
> > >
> > > | Quadruped Walk| 0.5h | 1.0h | 1.5h | 2.0h | 2.5h | 3.0h | 3.5h | 4.0h |
> > > | :--- | :---: | :---: | :---: | :---: | :---: | :---: | :---: | :---: |
> > > | DrQ-v2 (Base) | 147 | 290 | 364 | 715 | 732 | - | - | - |
> > > | DrQ-v2 (Widened) | 124 | 269 | 321 | 413 | 673 | 702 | 710 | 737 |
> > > | Ours (Full) | 191 | 327 | 413 | 737 | 756 | 783 | 804 | 840 |
> > > | Ours (Reduced) | 249 | 345 | 510 | 762 | 797 | 812 | 830 | - |
> > >
> > > | Walker Run| 0.5h | 1.0h | 1.5h | 2.0h | 2.5h | 3.0h | 3.5h | 4.0h |
> > > | :--- | :---: | :---: | :---: | :---: | :---: | :---: | :---: | :---: |
> > > | DrQ-v2 (Base) | 163 | 236 | 415 | 505 | 538 | - | - | - |
> > > | DrQ-v2 (Widened) | 161 | 195 | 301 | 392 | 472 | 501 | 525 | 558 |
> > > | Ours (Full) | 188 | 275 | 422 | 523 | 588 | 644 | 677 | 720 |
> > > | Ours (Reduced) | 204 | 333 | 463 | 579 | 642 | 699 | 721 | - |
> > >
> > > To strictly isolate architectural overhead, we report Train FLOPs as the total theoretical compute—including all corresponding forward and backward passes—for a single complete RL update step per sample.
> > >
> > > || Params | FLOPs | quadruped-walk (500k / 1M) | walker-run (500k / 1M) |
> > > | :--- | :--- | :--- | :--- | :--- |
> > > | DrQ-v2 (Base) | 7M | 404.4M | 343 ± 106 / 732 ± 91 | 436 ± 143 / 538 ± 115 |
> > > | DrQ-v2 (Widened) | 69M | 974.7M | 387 ± 71 / 760 ± 82 | 477 ± 62 / 558 ± 67 |
> > > | Ours (Reduced) | 18M | 363.7M | 726 ± 46 / 830 ± 54 | 496 ± 24 / 722 ± 10 |
> > > | Ours (Full) | 69M | 863.6M | 734 ± 31 / 846 ± 25 | 509 ± 43 / 721 ± 9 |
> > >
> > > As shown, the additional compute of our Full version remains within a manageable range. More importantly, our Reduced architecture effectively mitigates the expense.
> > >
> > > 2. Scalability to Large-scale Perception Tasks
> > >
> > > To further demonstrate our algorithm's efficiency in large-scale perception tasks with high-dimensional visual features, we conducted a scalability experiment in the CARLA autonomous driving environment. To handle complex visual inputs, the vision setup was expanded from a 3-camera to a 5-camera configuration.
> > >
> > > We evaluate the same four architectural configurations as in the DMControl experiments, scaled naturally to accommodate this high-dimensional setting:
> > >
> > > Widened DrQ-v2: As the massive 5-camera input intrinsically pushes the parameter count of Ours (Full) to ~382M, we further expanded the baseline's hidden dimension to 6144 (instead of 4608 used in DMControl) to strictly match this new parameter budget for a fair comparison.
> > >
> > > Ours (Reduced): This variant retains the exact same structural weight-sharing mechanisms detailed previously. It seamlessly adapts to the high-dimensional input, drastically optimizing computational efficiency.
> > >
> > > | | Default(200k) | HardRainNoon(200k) | Params | Train FLOPs |
> > > | :--- | :--- | :--- | :--- | :--- |
> > > | DrQ-v2 (Base) | 253.6 ± 36 | 264.1 ± 10 | 30.01M | 2.47G |
> > > | DrQ-v2 (Widened) | 261.2 ± 29 | 269.2 ± 7 | 381.6M | 6.22G |
> > > | Ours (Full) | 369.2 ± 18 | 364.3 ± 20 | 381.9M | 5.05G |
> > > | Ours (Reduced) | 363.1 ± 16 | 360.1 ± 15 | 43.94M | 2.68G |
> > >
> > > Results demonstrate that our algorithm effectively scales to high-dimensional visual inputs, delivering superior performance and training stability with only marginal computational overhead. This validates our architecture's exceptional efficiency in handling complex perception data for real-world autonomous environments.
> > >
> > > Note: In the CARLA experiments, wall-clock time is mainly limited by environment rendering and communication latency. The increase in computational overhead introduced by the algorithm is nearly negligible in actual training time. For this reason, we do not report wall-clock time results in the CARLA environment.
> > >
> > > We apologize for the delayed response, as we have been continuously running intensive scalability experiments to ensure the rigor of our results. We hope these additional data fully address your questions regarding the efficiency and scalability of our model.

---

### Decision · Program_Chairs · 2026-04-30

**Decision:**

Accept (regular)

**Comment:**

This paper introduces an effective multi-timescale critic framework for reinforcement learning that leverages a neural reward decoder and cross-attention fusion to dynamically balance short- and long-term planning objectives. While the reviewers initially raised valid concerns regarding parameter efficiency, hyperparameter sensitivity, and notational clarity, the authors supplied a highly convincing rebuttal featuring a capacity-matched reduced model and comprehensive ablation studies that successfully resolved the committee's reservations. Given the solid empirical results and the thoroughness of the authors' responses, my final recommendation is to accept the submission.